# Active presynaptic ribosomes in the mammalian brain, and altered transmitter release after protein synthesis inhibition

Matthew S Scarnati, Rahul Kataria, Mohana Biswas, Kenneth G Paradiso*

Department of Cell Biology and Neuroscience, Rutgers University, Piscataway, United States

**Abstract** Presynaptic neuronal activity requires the localization of thousands of proteins that are typically synthesized in the soma and transported to nerve terminals. Local translation for some dendritic proteins occurs, but local translation in mammalian presynaptic nerve terminals is difficult to demonstrate. Here, we show an essential ribosomal component, 5.8S rRNA, at a glutamatergic nerve terminal in the mammalian brain. We also show active translation in nerve terminals, in situ, in brain slices demonstrating ongoing presynaptic protein synthesis in the mammalian brain. Shortly after inhibiting translation, the presynaptic terminal exhibits increased spontaneous release, an increased paired pulse ratio, an increased vesicle replenishment rate during stimulation trains, and a reduced initial probability of release. The rise and decay rates of postsynaptic responses were not affected. We conclude that ongoing protein synthesis can limit excessive vesicle release which reduces the vesicle replenishment rate, thus conserving the energy required for maintaining synaptic transmission.

DOI: https://doi.org/10.7554/eLife.36697.001

## Introduction

Synaptic transmission requires the synthesis, localization, interaction and ongoing replenishment of thousands of pre- and postsynaptic proteins (*Witzmann et al., 2005*; *Gonzalez-Lozano et al., 2016*; *Loh et al., 2016*). The location and stoichiometry of each protein is highly regulated to maintain the necessary levels of precision and fidelity of signaling across the synapse. The highly structured and polarized morphology of neurons, with axons and dendrites that can project long distances, creates a unique challenge to maintain sufficient levels of numerous necessary proteins at distant locations (*Alvarez et al., 2000*; *Maday et al., 2014*; *Tasdemir-Yilmaz and Segal, 2016*). In addition, these remote regions need to rapidly modify the magnitude and duration of their responses, which can require changes in pre- and postsynaptic protein expression levels. Synaptic proteins are typically thought to be synthesized in the soma and transported to synapses, but several groups have demonstrated that some postsynaptic proteins can be synthesized locally in dendrites (*Pfeiffer and Huber, 2006*; *Jung et al., 2014*; *Rangaraju et al., 2017*). Over the past decade, RNA based mechanisms have been discovered that respond to extrinsic signals that affect postsynaptic local translation in dendrites, providing a mechanism to modify or maintain activity at specific regions (*Liu-Yesucevitz et al., 2011*; *Yoon et al., 2016*). This is possible due to the targeting of coding and non-coding RNA (*Vo et al., 2010*) with RNA binding proteins, and the presence of ribosomes that are located in, or moved to, specific neuronal regions or compartments (*Ostroff et al., 2002*). This allows the neuron to have the necessary components in place to translate specific dendritic proteins on-site, in response to specific signals. The role of local translation in resting and sustained levels of synaptic transmission is a major issue of interest.

*For correspondence:
kenparadisolab@gmail.com

Local protein synthesis is thought to provide a faster and more efficient mechanism for neurons to maintain or alter activity levels and respond to rapidly changing inputs. In mammalian central nervous system (CNS) neurons, local postsynaptic protein synthesis in dendrites is well established. In contrast, until recently, most evidence for local presynaptic protein synthesis in axons and nerve terminals came from invertebrates and the mammalian peripheral nervous system (*Alvarez et al., 2000*). Evidence for presynaptic protein synthesis in the mammalian brain has been difficult to demonstrate, largely due to the difficulties of accessing and imaging CNS presynaptic terminals (*Akins et al., 2009*). Despite these issues, presynaptic ribosomes have recently been shown to be present in GABA-ergic interneurons in the hippocampus of mature mice, where presynaptic protein synthesis is necessary to induce a long-term depression of synaptic responses (*Younts et al., 2016*). Local protein synthesis has also been shown to occur in the axons of developing mammalian brain neurons, and plays a role in establishing synapses (*Batista et al., 2017*) and affecting release at recently formed nerve terminals (*Taylor et al., 2013*). Although it is still highly debated, recent work provides good evidence that local protein synthesis can occur in nerve terminals in the mammalian brain, and it can affect presynaptic activity.

To better understand the role of presynaptic protein synthesis in the brain, we have used the calyx of Held synapse, located in the medial nucleus of the trapezoid body (MNTB) in the auditory brainstem (*von Gersdorff and Borst, 2002*). This synapse is involved in sound localization, and can maintain prolonged synaptic transmission at frequencies of 100 to 200 Hz. The calyx of Held is a large, glutamatergic nerve terminal that forms a monosynaptic, axosomatic connection onto MNTB principle cells. This large presynaptic terminal contains hundreds of individual release sites, and the size of the terminal facilitates imaging (*Rodríguez-Contreras et al., 2008*; *Körber et al., 2015*). In addition, the basic mechanisms of pre- and postsynaptic responses have been extensively characterized at this synapse (*Neher, 2017*). This synapse also undergoes significant developmental changes in its morphology and physiological characteristics that involves changes in presynaptic protein content, occurring around the onset of hearing in mice, at approximately postnatal day 10 (*Borst and Soria van Hoeve, 2012*). Finally, in a mouse brain, the calyx of Held nerve terminal is ~3 mm away from the cell body. This distance could enhance the need for local translation at the nerve terminal. These characteristics make this synapse an excellent model for studying the effects of local presynaptic protein synthesis on synaptic transmission.

Our data show that presynaptic ribosomes are present, functional, and active under non-stimulated conditions. In addition, we show that within ~1 hr of inhibiting protein synthesis, there is an increase in the frequency of spontaneous neurotransmitter release, an increase in the paired pulse ratio, and an increase in the amount of release throughout 100 Hz and 200 Hz stimulus trains. These findings demonstrate that local presynaptic protein synthesis occurs at the calyx of Held nerve terminal, and it affects basal levels of spontaneous neurotransmitter release as well as release during prolonged levels of evoked activity. This represents a previously unknown role for ongoing local translation in adjusting spontaneous and evoked vesicle release.

## Results

### Evidence for presynaptic ribosomes at the calyx of held nerve terminal

The calyx of Held is a large glutamatergic, monosynaptic nerve terminal located in the medial nucleus of the trapezoid body (MNTB) in the mammalian auditory brainstem (*Figure 1A*). Cell bodies in the anterior ventral cochlear nucleus (AVCN) project axons a significant distance to the MNTB, which is ~3 mm in a mouse brain (*Figure 1A*). Up to approximately postnatal day (P) 12, the calyx primarily has a spherical or spoon-shaped morphology (*Figure 1B*, left panels). This large spherical morphology provides a well-defined image of the presynaptic compartment that allows the ability to clearly distinguish fluorescent signals in the presynaptic terminal from fluorescence in the postsynaptic soma. By P12, the calyx terminal begins a change to a fenestrated morphology that is prevalent by P16, when it is considered to be mature in morphology and function (*Grande and Wang, 2011*) (*Figure 1B*, right panels). The morphological changes are accompanied by changes in protein expression that allow faster action potential kinetics (*Yang and Wang, 2006*) and synaptic release properties (*Borst and Soria van Hoeve, 2012*) that begin at ~P10, and allow this synapse to function at the high frequency and fidelity (*Taschenberger and von Gersdorff, 2000*) that is required for

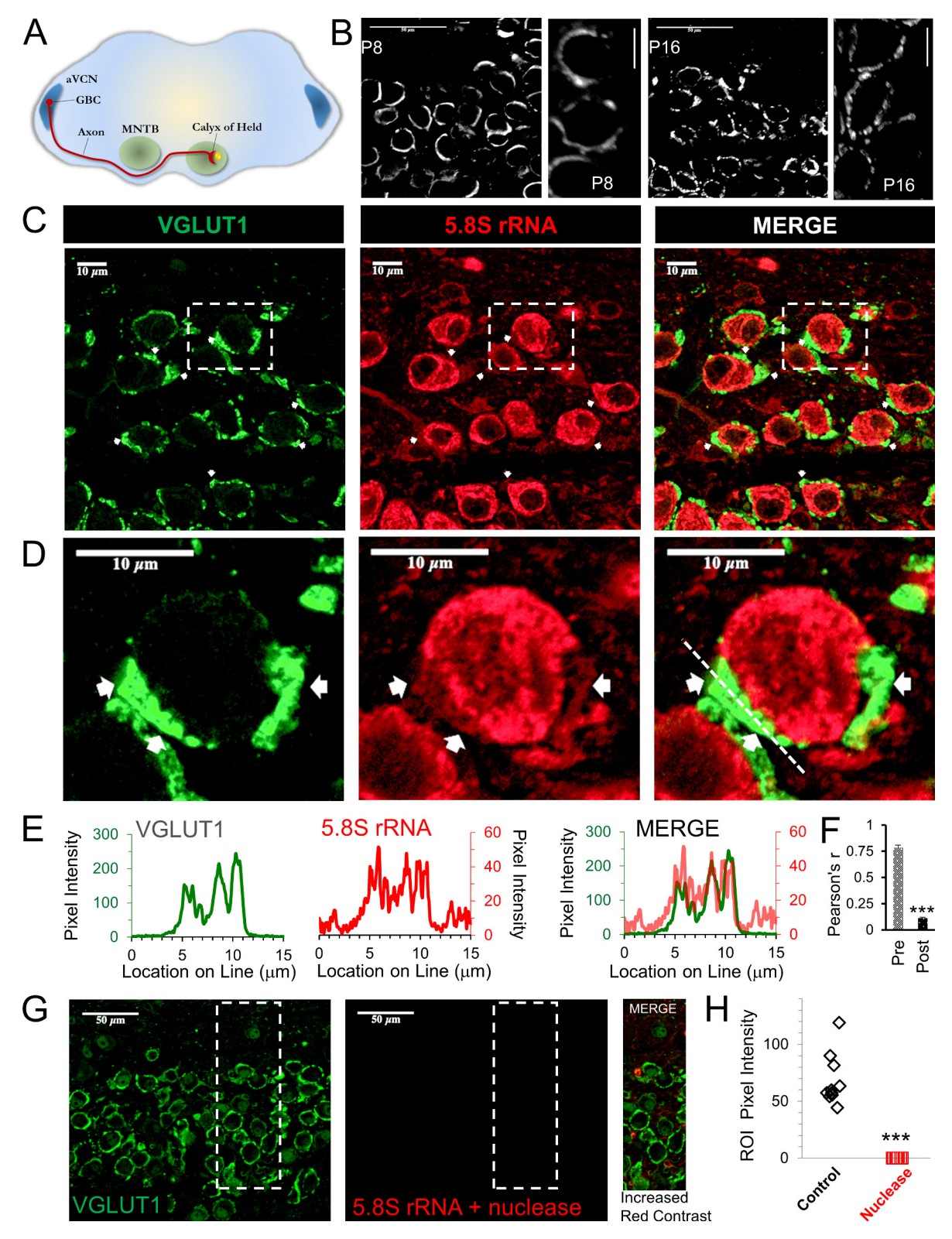

**Figure 1.** Presence of presynaptic ribosomes in brain slices. (**A**) Globular bushy cells (GBCs) in anteroventral cochlear nucleus (aVCN) project long axons (~3 mm) forming monosynaptic calyx synapses onto principal cells in the contralateral medial nucleus of the trapezoid body (MNTB). (**B**) VGLUT1 antibody labels synaptic vesicles in the calyx of Held nerve terminal. At postnatal day 8 (P8) the calyx largely surrounds the postsynaptic neuron. By P16, a fenestrated morphology with swellings is seen (scale: 50 μm or 10 μm). (**C**) VGLUT1 antibody labels synaptic vesicles in the calyx of Held nerve

*Figure 1 continued on next page*

*Figure 1 continued*

terminal. Immunolabeling of 5.8S rRNA identifies ribosomes. White arrows in each panel depict some examples of clear presynaptic 5.8S rRNA signal. (D) High optical zoom of dotted box in panel C. White arrows mark presynaptic 5.8S rRNA. (E) Line scan analysis depicts pixel intensity of VGLUT1 and 5.8S rRNA along a 15 μm line shown in *Figure 1D*. Merged line scans show excellent overlap in relative signal intensity of VGLUT1 and 5.8S rRNA. (F) Pearson's correlation coefficients (r) quantifies colocalization between the VGLUT1 and 5.8S rRNA signals, (n = 13 presynaptic terminals). The r-value of 0.78 ± 0.02, demonstrates high colocalization. The r-values for the matching postsynaptic cell body regions (n = 13) have an r-value of 0.11 ± 0.10 (p ≤ 0.001 t-test and KS2-test; AD-test for presynaptic dataset normal distribution, p = 0.06). (G) Nuclease treatment prior to 5.8S rRNA antibody binding eliminates the ribosomal signal. Enhanced contrast further shows the lack of ribosomal RNA after nuclease treatment. (H) Nuclease treatment effectively eliminates 5.8S rRNA signal compared to control conditions (p ≤ 0.001 t-test and KS2-test; AD-test control dataset normal distribution, p = 0.025).

DOI: https://doi.org/10.7554/eLife.36697.002
The following source data is available for figure 1:

**Source data 1.** 5.8s rRNAline scan, Pearson's r, and pixel intensity.
DOI: https://doi.org/10.7554/eLife.36697.003

sound localization (*Oertel, 1999*; *Carr et al., 2001*). Accordingly, there is a high level of protein turnover that occurs shortly before and throughout this period.

We hypothesized that local protein synthesis could occur at this nerve terminal, particularly due to the long axon length, its size (*Borst et al., 1995*) and high frequency firing (*Wu and Kelly, 1993*; *Borst et al., 1995*; *Taschenberger and von Gersdorff, 2000*) that requires high levels of proteins to maintain activity at the >600 release sites in a calyx nerve terminal (*Sätzler et al., 2002*; *Taschenberger et al., 2002*; *Wimmer et al., 2006*; *Dondzillo et al., 2010*). To investigate presynaptic protein synthesis, we first determined whether 5.8S ribosomal RNA (rRNA), a major component of the eukaryotic ribosome which is required to execute ribosomal translocation (*Lerner et al., 1981*; *Abou Elela and Nazar, 1997*; *Koenig et al., 2000*), is present in the nerve terminal. This component is eukaryotic specific and part of the large 80S ribosomal subunit. Therefore, this antibody does not label mitochondrial ribosomes which are more prokaryotic in their composition. This component has been shown to be present in dendritic compartments, axons (*Koenig et al., 2000*; *Zheng et al., 2001*; *Spillane et al., 2013*; *Taylor et al., 2013*) and neurites, as evidence for the presence of ribosomes (*Bolognani et al., 2004*; *Kim et al., 2005*; *Kim and Kim, 2006*; *Oyang et al., 2011*). Recent work, using super-resolution microscopy, has shown that 5.8S rRNA is present in nerve terminals of CA1 inhibitory interneurons in mice (*Younts et al., 2016*). Given the small size of most presynaptic terminals, standard imaging techniques can be difficult. Therefore, the large size of the calyx of Held nerve terminal helps to determine the presence and localization of 5.8S rRNA in the nerve terminal.

To label the large calyx of Held nerve terminal, an antibody against the vesicular glutamate transporter, VGLUT1, was used (*Figure 1C,D*; left panel, green), which labels most of the presynaptic compartment by marking glutamatergic synaptic vesicles (*Billups, 2005*; *Rodríguez-Contreras et al., 2008*; *Fioravante et al., 2011*; *Chen et al., 2013*; *Kempf et al., 2013*). Immunolabeling for 5.8S rRNA in brain slices shows a robust signal (*Figure 1C,D*; center panel, red), particularly in neuronal somata, consistent with the high levels of protein synthesis that occur in the cell body (*Palay and Palade, 1955*; *Giuditta et al., 2008*). We typically observe several areas that exhibit clear 5.8S rRNA labeling in the presynaptic terminal, as shown in the representative images (*Figure 1C,D*). In the region shown here, the average intensity ratio for the presynaptic terminal to background signal is 3:1, which allowed us to clearly distinguish the presynaptic signal. As expected, the average intensity for the signal in the presynaptic terminal was less than that of the somata, with an average intensity ratio of 4:1 for somata to presynaptic signals. Despite the strong postsynaptic fluorescence, we were able to unambiguously identify numerous areas with clear presynaptic signals (see white arrows in *Figure 1C and D*). We note that there are also numerous areas where a presynaptic signal is present, but background fluorescence from adjacent neuronal and glial cells made it more difficult to show the specificity of the presynaptic signal (*Figure 1C*). To determine the overlap of the 5.8S signal with the VGLUT1 signal, we graphed the fluorescence intensity along a line that included the presynaptic terminal plus adjacent areas (*Figure 1E*) and compared the positional overlap of VGLUT1 and 5.8S rRNA intensities at high magnification (*Figure 1D*; line shown in right panel). We find a high correlation between the relative intensities of the 5.8S rRNA and VGLUT1 signals

(*Figure 1E*, merge), with similar peaks and troughs in their intensities. The troughs in signal intensity could be due to the presence of organelles and other presynaptic components that reduce both signals. Finally, we performed correlation-coefficient analysis to quantify the overlap between VGLUT1 and 5.8S rRNA. We calculated a Pearson's correlation-coefficient (r) of 0.78 ± 0.04 SEM for VGLUT1 and 5.8S rRNA signal in the presynaptic terminal. For comparison, and to control for the possibility of background fluorescence, we calculated a Pearson's r-value of 0.11 ± 0.01 SEM in the region surrounded by the VGLUT1 signal, corresponding to the somatic compartment of the postsynaptic neuron (*Figure 1F*, n = 13 neuronal pairs, p < 0.001; t-test and KS2-test). This provides very strong evidence that ribosomes are present in the calyx of Held presynaptic nerve terminal.

To verify the specificity of the 5.8S rRNA signal, we treated the brain slices with nucleases to degrade the 5.8S rRNA prior to antibody labeling and found this eliminated the presynaptic and postsynaptic 5.8S signal (*Figure 1G*, center). To better visualize any residual 5.8S signal remaining after nuclease treatment, we maximized the signal contrast (*Figure 1G*, right panel), but still found a complete lack of 5.8S signal in nerve terminals and somata. The average pixel intensity for the combined pre- and postsynaptic compartments in untreated slices (68.30 ± 7.33 SEM, n = 10) was far greater than the low remaining signal in nuclease treated slices (0.12 ± 0.01 SEM, n = 10) from the same brain (*Figure 1H*). These data further demonstrate the presence of ribosomes in the presynaptic terminal, suggesting the ability for local protein synthesis at this nerve terminal.

## Functional presynaptic ribosomes

Our data show that a major ribosomal component is present in the presynaptic terminal. To determine if these ribosomes are fully assembled and functional, we used the SUrface SEnsing of Translation (SUnSET) technique. This technique allows us to directly visualize locations of active protein synthesis using a fluorescent signal that is proportional to the amount of translation. Briefly, this method uses puromycin, which mimics tRNA and becomes incorporated into nascent polypeptide chains. Fixation, followed by specific antibody labeling detects the amount and location of polypeptides that have incorporated puromycin, which demonstrates the presence of active ribosomes (*Schmidt et al., 2009*; *Goodman et al., 2012*; *Goodman and Hornberger, 2013*). As described below, our results using this technique confirm that functional ribosomes are present in the presynaptic terminal.

A 10 min application of puromycin allowed us to detect ribosome activity in brain slices. We found fluorescent signal in calyx of Held nerve terminals and in principal cell somata, with relatively low background activity from other nearby cells (*Figure 2A*). We note that there are regions with high levels of ribosomal activity in adjacent neurons and glia (*Figure 2A* and *Figure 2—figure supplement 1A,B*). To illustrate presynaptic ribosomal activity, we highlight one of the regions where the activity from adjacent cells was particularly low, thus allowing us to clearly demonstrate the presence of a presynaptic signal for ribosomal activity (*Figure 2A,F,G*). To quantify the presence of active ribosomes in the presynaptic compartment, we calculated the Pearson's correlation coefficient (r) for the puromycin and VGLUT1 signals. We find r-values of 0.74 (±0.05 SEM) for the presynaptic terminal compared to 0.14 (±0.03 SEM) for the somatic region surrounded by the calyx nerve terminal (*Figure 2B*, n = 13 neurons, p < 0.001 t-test and KS2-test). The postsynaptic neuron is typically intact, but it can be damaged during slice preparation. Interestingly, we also find a puromycin signal at some presynaptic terminals where the signal from the postsynaptic neuron is low or absent, likely due to damage or disintegration of the postsynaptic neuron during slice preparation (*Figure 2C*). This serves to further demonstrate an isolated presynaptic signal. Localizing puromycin fluorescence to the presynaptic terminal demonstrates the presence of active presynaptic ribosomes. As expected, the fluorescence intensity is higher in somata than in the terminal. At a higher magnification, puromycin labeling is clearly visible in the presynaptic terminal demonstrating the presence of functional presynaptic ribosomes (*Figure 2F,G*). To better visualize the relationship between puromycin labeling and the presynaptic marker VGLUT1, we used line scans to assess the degree of colocalization (*Figure 2*, F, G graphs). We find excellent agreement in the location and relative intensity of the two signals (*Figure 2F,G*, graphs on right showing merged line scan overlay), thus demonstrating that translationally competent ribosomes are found in the calyx of Held presynaptic nerve terminal, and they can be active under resting conditions. We note that longer puromycin application periods were tested, and this produced a strong pre- and postsynaptic signal, but longer application times also sharply increased the background signal, presumably due to activity from other

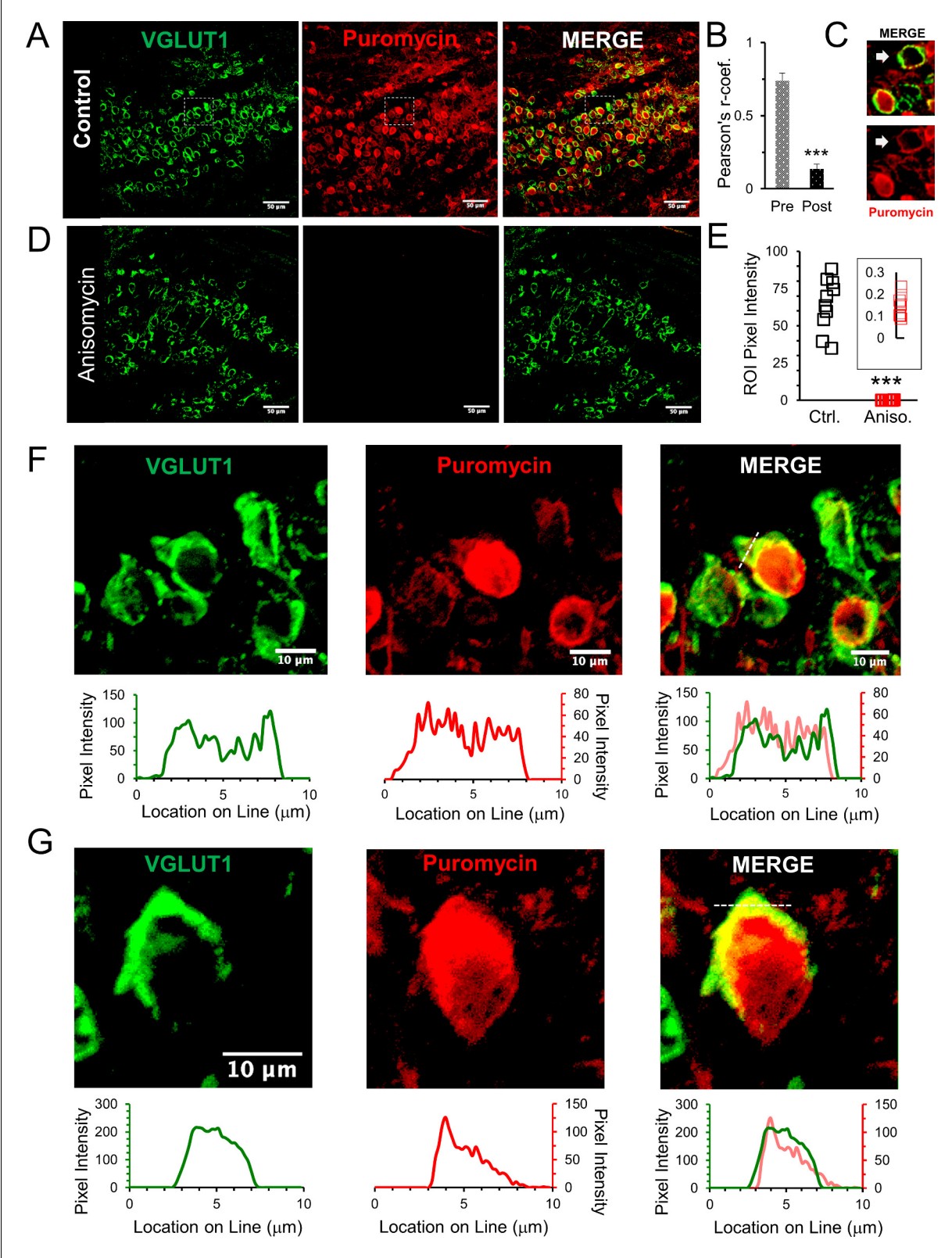

**Figure 2.** Active ribosomes in the presynaptic terminal in brain slices. (**A**) VGLUT1 antibody identifies the presynaptic terminal by labeling synaptic vesicles in the calyx of Held nerve terminal. Puromycin, tRNA analog, labeling shows active translation sites (SUnSET). Merge VGLUT1 +Puromycin shows active presynaptic ribosomes. (**B**) Pearson's correlation coefficient (**r**) quantifies colocalization between VGLUT1 and puromycin signals for presynaptic terminals (n = 13). Average r = 0.74 ± 0.05, demonstrates high colocalization. A comparison, average r in corresponding postsynaptic cell
*Figure 2 continued on next page*

*Figure 2 continued*

bodies = 0.14 ± 0.03 (p ≤ 0.001 t-test and KS2-test; AD-test presynaptic dataset normal distribution, p = 0.06, AD-test postsynaptic dataset normal distribution, p = 0.05; n = 13). (**C**) Example of presynaptic puromycin signal (arrow) in the absence of a postsynaptic signal. Likely due to damage and deterioration of postsynaptic neuron with intact terminal. (**D**) Application of translational inhibitor (anisomycin, 40 µM) for 1 hr prior to puromycin treatment eliminates puromycin labeling (center panel and right panel), showing specificity of puromycin binding to active ribosomes. (**E**) Anisomycin treatment eliminates the puromycin signal. Inset shows higher detail of the residual ROI pixel intensity following anisomycin treatment (Pixel Intensity Ctrl = 64.4 ± 1.76, Aniso = 0.14 ± 0.02; p ≤ 0.001 t-test and KS2-test; n = 10). (**F**) Top Panels: higher optical zoom (63x) of dotted box in panel 2A. Bottom Panels: Line scan analysis shows pixel intensity of VGLUT1 and puromycin along the 10 µm line shown in merged image (right panel). Line scan overlay shows a high overlap in relative signal intensity of VGLUT1 and puromycin along scan line. (**G**) Additional example from a different brain slice (PN8) shows active ribosomes in both the pre- and postsynaptic compartments, demonstrated in line scans below each image. Merged overlay shows an excellent match in relative intensities of VGLUT1 and puromycin.

DOI: https://doi.org/10.7554/eLife.36697.004

The following source data and figure supplement are available for figure 2:

**Source data 1.** Puromycin line scans, Pearson's r, and pixel intensity.
DOI: https://doi.org/10.7554/eLife.36697.006
**Figure supplement 1.** Active ribosomes adjacent to the calyx of Held in brain slices.
DOI: https://doi.org/10.7554/eLife.36697.005

neurons and glia in the slice (data not shown). We conclude that all of the necessary components (mRNA, tRNA, rRNA, ribosomal proteins and ribosomal binding proteins), which are required to execute translation, must also be located in the nerve terminal.

To validate that the SUnSET assay detects active translation, we treated brain slices with the translational inhibitor anisomycin for 1 hr prior to the addition of puromycin (*Figure 2D*). Ribosomes must be active for puromycin to be incorporated into a polypeptide chain to be detected by the SUnSET assay. Consistent with this, we find that anisomycin treatment effectively eliminates the puromycin signal (*Figure 2D*, center), giving a > 100 fold reduction in the fluorescence intensity following anisomycin treatment (*Figure 2E*). An important point is that anisomycin specifically inhibits eukaryotic translation. Although it is unclear if mitochondrial protein synthesis can be measured using the SUnSET assay, any potential signal from mitochondrial protein synthesis would still be present after anisomycin treatment. We note that anisomycin has been used to inhibit cellular protein synthesis in mammalian cells in order to study mitochondrial protein synthesis (*Richter-Dennerlein et al., 2016*). We conclude that the complete loss of SUnSET signal that was induced by anisomycin treatment shows that mitochondrial protein synthesis does not contribute to the active ribosome signal we see here. These results demonstrate that the SUnSET assay provides an efficient and specific measurement of the presence and general location of active ribosomes. This verifies that the necessary components for protein synthesis are present in the presynaptic terminal and are capable of forming translationally competent ribosomes that are active.

## Spontaneous synaptic events indicate presynaptic effects of inhibiting translation

To determine if ongoing protein synthesis affects synaptic transmission, we first looked at miniature excitatory postsynaptic currents (mEPSCs) to determine if spontaneous release events are affected after protein synthesis is inhibited. The frequency of spontaneous events is due to presynaptic release properties (*Kavalali, 2015*) while the amplitude and shape of the response are largely attributed to postsynaptic changes in ionotropic receptor responses. However, presynaptic properties such as the level of neurotransmitter filling in vesicles can also affect the amplitude of the postsynaptic current (*Goh et al., 2011*). Since the need for protein synthesis could be affected by prior activity, it was important to measure spontaneous activity at several times during our recordings to determine if inhibiting translation affects the initial mEPSCs, and the mEPSCs that occur after evoked activity. As noted in the Materials and methods section, application of the protein synthesis inhibitor in the physiology experiments, and the subsequent data analysis, were performed blinded.

We find that the initial mEPSC frequency, measured shortly after onset of whole-cell recording, is higher in cells treated with the protein synthesis inhibitor anisomycin (2.4 ± 0.7 Hz, n = 11 cells) compared to untreated neurons (1.4 ± 0.3 Hz, n = 10 cells) from the same animals (*Figure 3A*). The initial cumulative probability histogram of the time between mEPSC events clearly shows that inhibiting

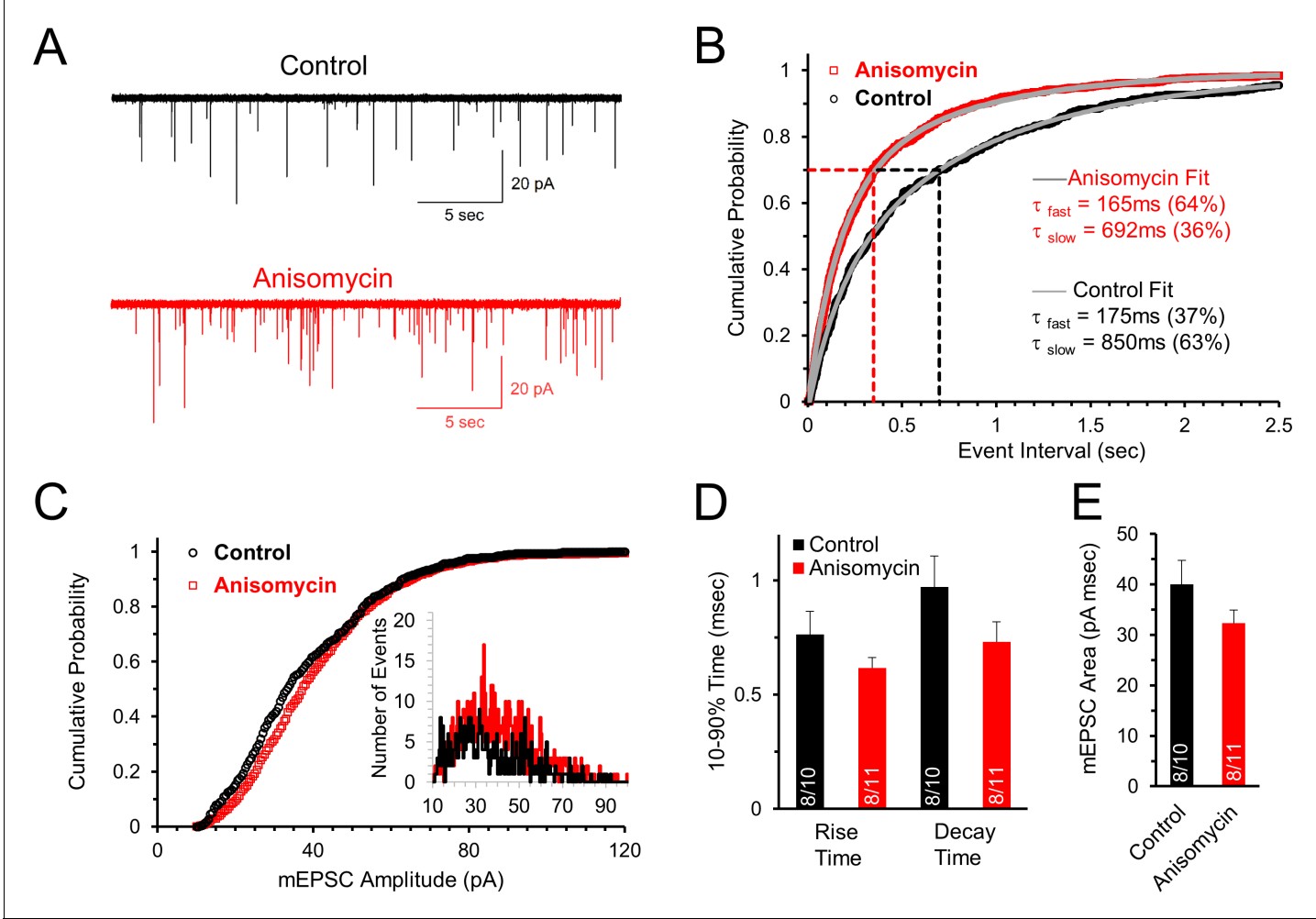

**Figure 3.** Initial spontaneous release frequency is higher in neurons treated with protein synthesis inhibitor, demonstrating a presynaptic effect. (**A**) Representative recordings of the initial spontaneous release in control neurons (black) and neurons treated with the translational inhibitor anisomycin (red). (**B**) Cumulative probability of the intervals between spontaneous release events measured by mEPSCs in neurons treated with protein synthesis inhibitor (red) and control neurons (black). The time constants and amplitudes of double exponential fits (gray versus blue lines) show a reversal in the percent contributions of the fast and slow release components for the population of event intervals in control and translation inhibited neurons. (**C**) Cumulative probability of mEPSC amplitudes in protein synthesis inhibited (red) and control (black) neurons. Inset histogram: number of release events at different amplitudes. (**D**) Average mEPSC 10–90% rise time for control and anisomycin treated neurons (left; control = 0.76 ± 0.1; aniso = 0.62 ± 0.05) and decay time (right; control = 0.97 ± 0.14; aniso = 0.73 ± 0.09). Bar graph n-values: # of animals/ # of neurons. (**E**) Average mEPSC area for control (40 ± 4.7) and anisomycin treated neurons (32.4 ± 2.6).

DOI: https://doi.org/10.7554/eLife.36697.007

The following source data is available for figure 3:

**Source data 1.** Initial spontaneous release following anisomycin treatment.
DOI: https://doi.org/10.7554/eLife.36697.008

protein synthesis decreases the time between mEPSC events compared to control recordings of neurons in untreated slices from the same animals (*Figure 3B*, p < 0.001, Kolmogorov-Smirnov test). For example, in neurons treated with anisomycin, 70 percent of the events occur with an interval less than ~350 msec (*Figure 3B*, red dotted line), compared to ~700 msec for control recordings (*Figure 3B*, black dotted line). To further examine this, we fit exponential curves to the cumulative probability data to quantify the time course of spontaneous release. We find that both the control and the protein synthesis inhibited cumulative probability distributions are well fit by double exponential curves, indicating a population of short inter-event intervals and a population of longer inter-event intervals between spontaneous release responses (*Figure 3B*, grey lines through data points).

In the exponential fits for both the control and anisomycin-treated neurons we find a fast component with a time constant of ~170 msec, and a second component which is >4 fold slower (*Figure 3B*). Interestingly, the fast component of the exponential fit, which corresponds to shorter event intervals, accounts for the majority of mEPSC events for anisomycin treated neurons (64%), in contrast to control neurons where the brief event intervals only account for 37% of the mEPSC event intervals. Thus, the percentage of brief versus longer intervals between spontaneous release events were equal but opposite in their distribution. Therefore, inhibiting protein synthesis causes an increase in the frequency of spontaneous release events.

In contrast to the differences seen in the frequency of mEPSCs, the amplitudes of mEPSCs were similar for control (36.8 ± 2.5 pA, n = 10 cells) and protein synthesis inhibited neurons (38.7 ± 2.0 pA, n = 11 cells), as shown in the cumulative probability of the mEPSC amplitudes (*Figure 3C*) and the amplitude histogram (*Figure 3C*, inset). For the shape of the mEPSCs, the protein synthesis inhibited neurons have a similar rise time (*Figure 3D*, p = 0.19 t-test; 0.57 KS2-test; AD-test control data normal distribution p = 0.06), decay time (*Figure 3D*, p = 0.14 t-test; 0.54 KS2-test), and average mEPSC area (*Figure 3E*, p = 0.16 t-test; 0.29 KS2-test) with no statistically significant differences between them. In summary, the absence of an effect on the mEPSC amplitude, and small, non-significant effects on the shape of the mEPSC, demonstrate that the postsynaptic response is not significantly affected by inhibiting protein synthesis for ~1–2 hr. However, the differences in the mEPSC frequency demonstrate that inhibiting protein synthesis has a presynaptic effect on the probability of spontaneous release.

## Enhanced spontaneous release following tetanus eliminates mEPSC frequency differences between control and protein synthesis inhibited neurons

Prolonged stimulation produces a transient elevation in the frequency of spontaneous release events (*Habets and Borst, 2006*). Given our finding that inhibiting protein synthesis also increases the frequency of spontaneous release events, we determined if the two effects act independently. Accordingly, we measured the frequency of spontaneous release in control and protein synthesis inhibited neurons, before and shortly after a tetanic stimulation at 100 Hz for 4 sec. It is important to note that prior to the pre-tetanus mEPSC recording, the neurons had received several rounds of evoked activity which is discussed in the next section. Although there was a ≥2-min period without evoked stimulation to allow recovery for the pre-tetanus recording (*Figure 4A$_1$*), there is still a small increase in the mEPSC frequency in both control (2.5 ± 0.45 Hz) and protein synthesis inhibited neurons (3.6 ± 0.89 Hz) compared to the spontaneous frequency measured before any evoked responses were given (*Figure 3A*). The difference in the timing of mEPSC events in control and protein synthesis inhibited neurons prior to tetanic stimulation is still clearly visible in the cumulative probability histogram of mEPSC event intervals (*Figure 4B$_1$*; p < 0.001, Kolmogorov-Smirnov test). Furthermore, both the protein synthesis inhibited, and control neurons continue to show a fast and slow process for spontaneous release, with the fast component accounting for the majority of release event intervals in protein synthesis inhibited neurons ($\tau_{fast}$ = 143 msec, 70%; $\tau_{slow}$ = 512 msec) and the minority of release event intervals in control neurons ($\tau_{fast}$ = 143 msec, 27%; $\tau_{slow}$ = 400 msec). Next, we delivered a tetanic stimulation (100 Hz, 4 sec) and measured the mEPSC frequency starting <5 sec after tetanic stimulation. Interestingly, following tetanic stimulation, the frequency of spontaneous release is nearly identical for both control (6.1 ± 0.86 Hz) and protein synthesis inhibited (6.9 ± 1.2 Hz) neurons (*Figure 4A$_2$*). This is best shown in the cumulative probability histogram of the mEPSC intervals, where the control and protein synthesis inhibited mEPSC curves partially overlap, and no longer have a statistically significant difference (*Figure 4B$_2$*, p = 0.35, Kolmogorov-Smirnov test). Finally, although a population of brief event intervals and longer event intervals are still present following tetanic stimulation (*Figure 4B$_2$*), there was a decrease in the average brief event interval ($\tau \cong$ 75-100 msec) and a corresponding decrease in the duration of longer event intervals ($\tau \cong$ 250-300 msec) following tetanic stimulation for both control and protein synthesis inhibited neurons. Furthermore, following tetanic stimulation, brief event intervals accounted for approximately 70% of the spontaneous release events for both control and protein synthesis inhibited neurons. The finding that the cumulative probability of the release intervals overlap following tetanic stimulation demonstrates that the effects of tetanic stimulation are greater for the control conditions. The smaller relative effect of tetanic stimulation after anisomycin treatment could indicate that protein synthesis

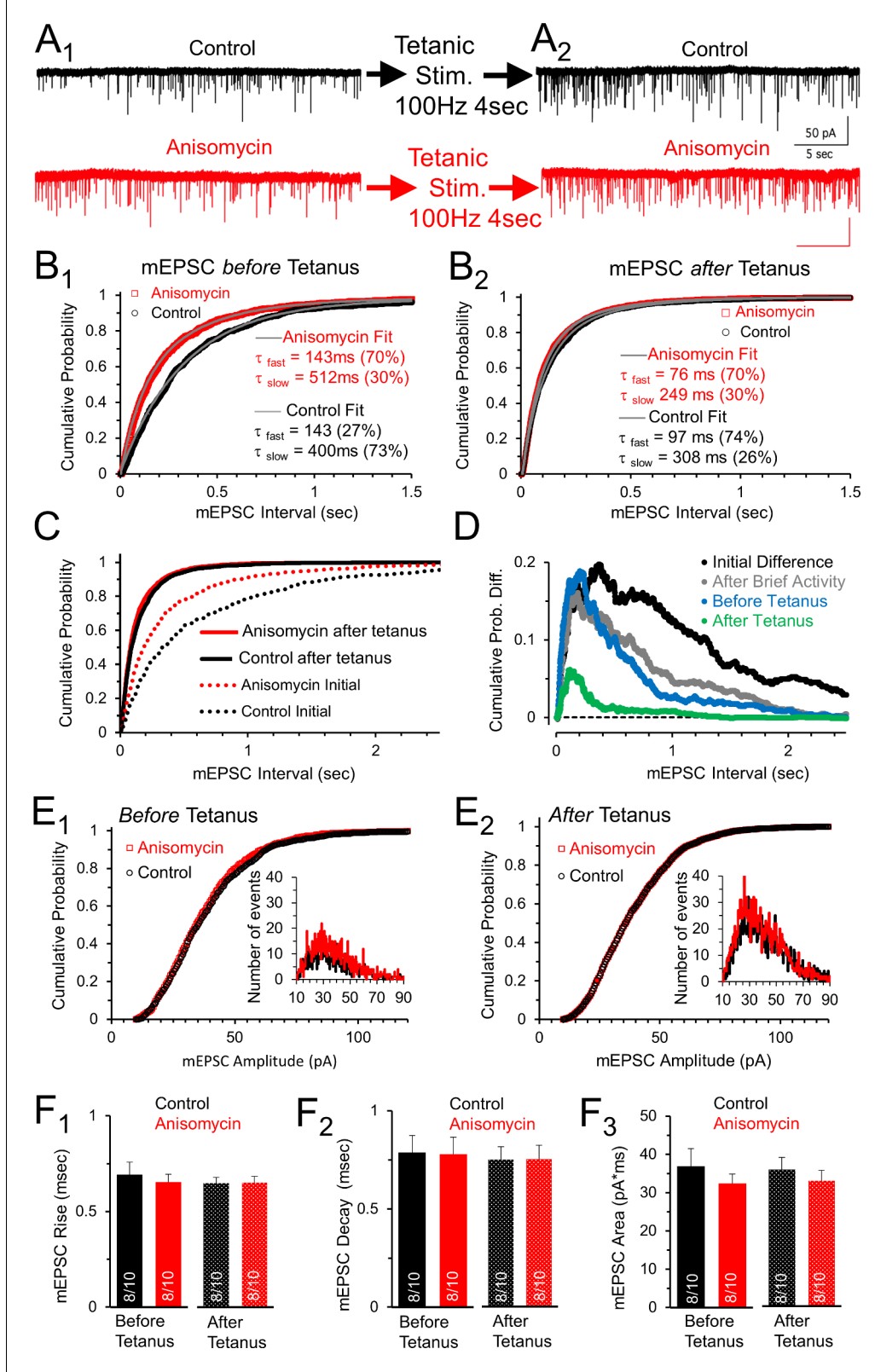

**Figure 4.** Spontaneous release following tetanus eliminates differences in mEPSC frequency between control and protein synthesis inhibited neurons. ($A_1$) Spontaneous release in control (black) and slices treated with the translational inhibitor anisomycin (red). Earlier activity (0.4 s, 100 and 200 Hz, and 4 s 100 Hz) increased frequency in both groups. ($A_2$) Spontaneous release in control (black traces) and neurons treated with the protein synthesis inhibitor anisomycin (red) following tetanic stimuli. ($B_1$) Cumulative probability of interval times between spontaneous release events in neurons treated

*Figure 4 continued on next page*

*Figure 4 continued*

with anisomycin (red) and control neurons (black) before tetanic stimulation. Double exponential fits (gray) show differences in the percent contributions of the fast and slow components. ($B_2$) Cumulative probability of intervals between mEPSC events after a tetanic stimulation. Double exponential fits (gray) show percent differences in the fast and slow components. (C) Initial cumulative probability of release (dotted lines) compared to release following tetanic stimulation in control (black) and neurons treated with anisomycin. (D) Differences in cumulative probability of release in anisomycin treated minus control neurons: initial difference (black); after brief activity (100 and 200 Hz for 400 msec each, gray); before tetanic stimulation (blue); following tetanic stimulation (green). ($E_1$) Cumulative probability of mEPSC amplitudes in protein synthesis inhibited (red) and control (black) neurons prior to evoked stimulation. Inset: depicts histogram of the mEPSC amplitudes. ($E_2$) Cumulative probability of mEPSC amplitudes in protein synthesis inhibited neurons (red) and control (black) following a tetanic stimulation. Inset: mEPSC amplitude histogram. ($F_1$) mEPSC 10–90% rise time for control (black) and anisomycin (red) before (left, $p = 0.62$ t-test and KS2-test; AD-test for normal distribution control $p = 0.04$) and following tetanic stimulation (right; $p = 0.96$ t-test; $p = 1$ KS2-test; n-values in bar graph apply to all panels Figure F: 8 animals and 10 neurons). ($F_2$) Average mEPSC 10–90% decay time for control (black) and neurons treated with anisomycin (red) before (left; $p = 0.94$ t-test; $p = 0.97$ KS2-test) and following tetanic stimulation (right). ($F_3$) Average mEPSC area for control (black) and neurons treated with anisomycin (red) before (left; $p = 0.39$ t-test; $p = 0.67$ KS2-test) and following tetanic stimulation (right; $p = 0.48$ t-test; $p = 0.67$ KS2-test).

DOI: https://doi.org/10.7554/eLife.36697.009
The following source data is available for figure 4:

**Source data 1.** Anisomycin effects on spontaneous release, before and after tetanus.
DOI: https://doi.org/10.7554/eLife.36697.010

inhibition and tetanic stimulation have similar presynaptic mechanisms that act to increase the frequency of spontaneous release events.

Comparing the initial mEPSC intervals present at the onset of each recording, before evoked stimulation (*Figure 3B*), with the intervals following tetanic stimulation (*Figure 4B_2*) shows that tetanic stimulation increases the fast and slow rates of release, and the percentage of rapid spontaneous release events (*Figure 4C*). To further show how spontaneous release differs between protein synthesis inhibited neurons and control neurons, we took the initial cumulative probability distributions (*Figure 3B*) and subtracted the average values in control neurons from the average values in the anisomycin treated neurons (*Figure 4D*, black trace). We also show how activity affects this difference. Previous activity reduces the difference in the distribution of the release events between protein synthesis inhibited and control neurons (*Figure 4D*, black versus grey lines), and a tetanic stimulation nearly eliminates these differences (*Figure 4D*, blue versus green lines). Therefore, the presynaptic effects of evoked activity act to speed up the rate of spontaneous release in control and protein synthesis inhibited neurons. In addition, a tetanic stimulation further increases the rate of release for control and inhibited neurons, and temporarily alters the control neurons so that the majority of spontaneous events occur by the fast component of release.

In contrast to the changes in spontaneous release, the amplitudes of the mEPSCs were unaffected by anisomycin treatment, tetanic stimulation or both combined (*Figure 4E1 and E2*). Furthermore, the mEPSC rise times (*Figure 4F_1*; $p = 0.62$ t-test and KS2 test), decay times (*Figure 4F_2*; $p = 0.95$ t-test; $p = 0.97$ KS2 test), and area (*Figure 4F_3*; $p = 0.39$ t-test; $0.67$ KS2 test) were also not affected by treatment with anisomycin, or by tetanic stimulation, or by both combined (for each tested pair: $p > 0.42$ t-test; $p > 0.67$ KS2-test). Collectively, these results show that inhibiting protein synthesis and presynaptic tetanic stimulation both have little to no effect on the amplitude and shape of the postsynaptic currents generated by spontaneous release. Therefore, the effects of inhibiting protein synthesis, and the effects of tetanic stimulation on mEPSC properties are specific to presynaptic effects on the rate of spontaneous release.

## EPSC kinetics are not affected when translation is inhibited for 1–2 hr

The finding that spontaneous release of synaptic vesicles is affected by inhibiting protein synthesis suggests that evoked responses could also be affected. Accordingly, we measured the effects of inhibiting protein synthesis on the response to stimulation at 100 and 200 Hz. First, prior to high frequency stimulation, at the beginning of each recording, we stimulated at a low frequency (0.1 Hz) to assess the initial quality of the recording (see Materials and methods) and determine if the peak amplitude, shape, and latency of evoked excitatory postsynaptic currents (EPSCs) are affected when protein synthesis is inhibited. In agreement with the results from the mEPSC measurements, the shape of the EPSCs appears to be unaffected by protein synthesis over the time course of 45 to 120

min of inhibition (*Figure 5A*). The 10–90% rise and decay times, and the latency time for control and for protein synthesis inhibited neurons were similar (p > 0.15; *Figure 5A–D*). However, there is a decrease in the peak amplitude of the initial evoked responses (*Figure 5E*) in control (6.21 ± 0.41 pA, n = 8) and anisomycin treated cells (4.05 ± 0.64 pA; n = 8; p = 0.013 t-test; p = 0.092 KS2-test). While the t-test indicates a statistically significant difference, we note that the KS2 test result did not. However, consistent with a reduction in the peak response, we find a difference in the area of

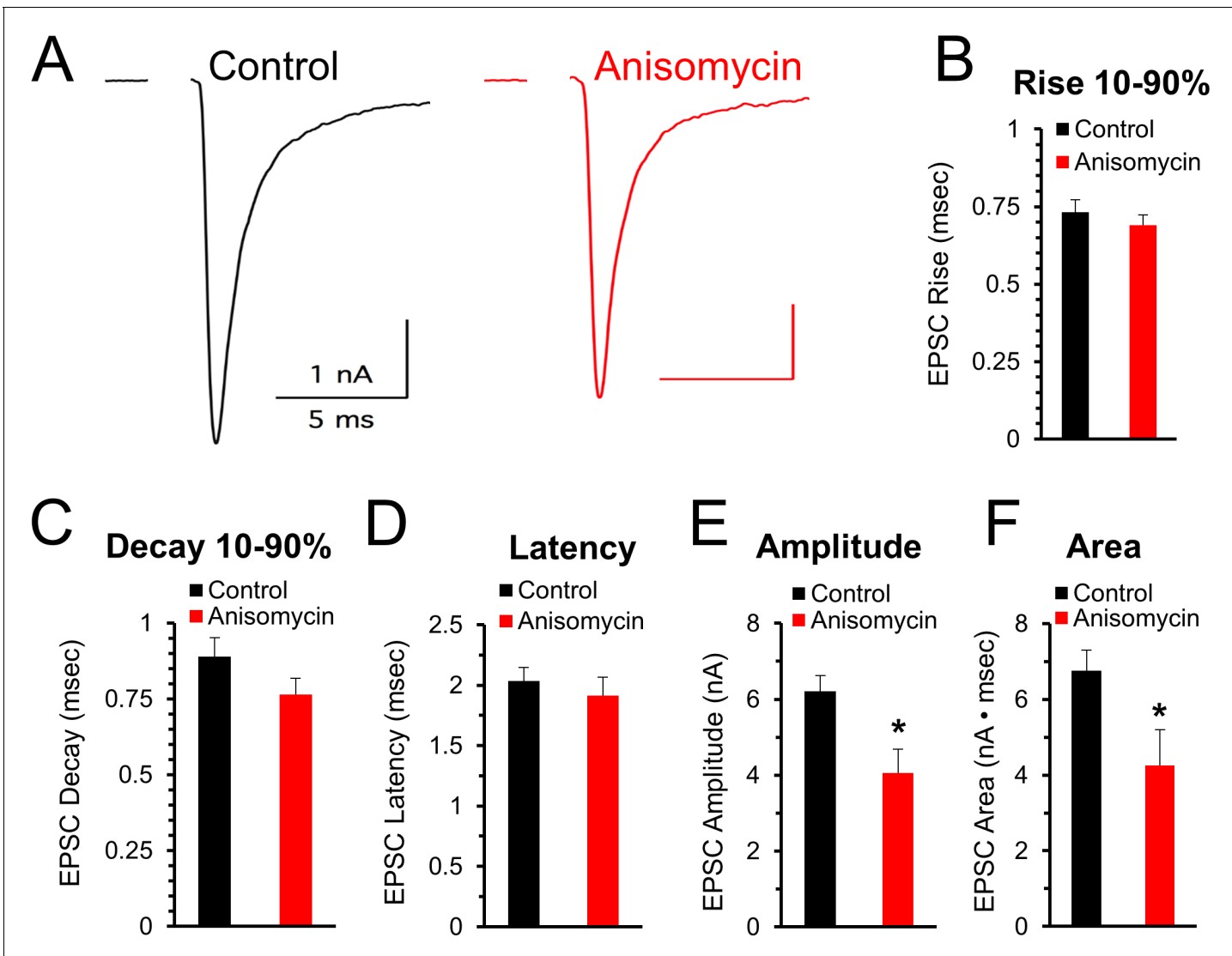

**Figure 5.** Initial evoked response timing and shape are not affected by inhibiting protein synthesis. (A) Representative traces of evoked excitatory postsynaptic currents (EPSCs) for control (black) and neurons treated with anisomycin (red) from PN10 mice. For both traces, horizontal scale bar is 5 msec, vertical scale bar is 1 nA. (B) Average 10–90% evoked EPSC rise time for control (black) and anisomycin (red) treated neurons (p = 0.43 t-test; 0.96 KS2-test; n = 9 control, eight anisomycin). (C) Average 10–90% evoked EPSC decay time for control (black) and anisomycin (red) treated neurons (p = 0.15 t-test; 0.28 KS2-test; n = 9 control, eight anisomycin). (D) Average evoked EPSC latency for control (black) and anisomycin (red) treated neurons. (p = 0.52 t-test; 0.69 KS2-test; n = 9 control, eight anisomycin). (E) Average evoked EPSC amplitude, control (black) and anisomycin (red) treated neurons. (p = 0.01 t-test; 0.09 KS2-test; n = 9 control, eight anisomycin). (F) Average evoked EPSC area, control (black) and anisomycin (red) treated neurons (p = 0.02 t-test; 0.007 KS2-test; AD-test for normal distribution control p = 0.027; n = 9 control, eight anisomycin).

DOI: https://doi.org/10.7554/eLife.36697.011

The following source data is available for figure 5:

**Source data 1.** Initial evoked release properties for anisomycin.
DOI: https://doi.org/10.7554/eLife.36697.012

the response (*Figure 5F*) in control (6.75 ± 0.54 nA·msec, n = 8) and anisomycin treated cells (4.26 ± 0.94 pA; n = 8; p = 0.04; p = 0.01 KS2-test). Taken together, these data indicate that the kinetics of the initial evoked postsynaptic responses are unaffected by inhibiting translation over a 1- to 2 hr time course, which is similar to our finding that mEPSC shape is not affected by inhibiting translation. However, the difference in the size and area of the EPSCs from control versus anisomycin treated neurons indicates that evoked neurotransmitter release appears to be affected by inhibiting protein synthesis. To further pursue this, in this same set of recordings, we also tested if inhibiting protein synthesis affects the EPSC responses at high frequency stimulation.

## Paired pulse measurements indicate a presynaptic effect of translational inhibition

The calyx of Held can fire at high frequencies with a high level of precision. We hypothesized that presynaptic protein synthesis may play a role in presynaptic mechanisms of synaptic transmission. We note that high frequency stimulation recruits additional presynaptic components that could be affected by inhibiting protein synthesis. Therefore, we tested short stimulus trains at 200 Hz for 400 msec which were followed two or more minutes later by a stimulus at 100 Hz. At both frequencies, we observed a lower level of depression in the responses from translation inhibited neurons. To quantify this, we first measured the effects of high frequency stimulation on the paired pulse ratio at 200 Hz, measured as the second EPSC (P2) response divided by the first EPSC (P1) response (*Figure 6A1and A2*). We find paired pulse depression at an interpulse interval (IPI) of 5 msec in control cells (0.72 ± 0.07 SEM, n = 9 cells from seven animals) but a facilitation in translation inhibited cells at the same interval (1.09 ± 0.09 SEM; p = 0.004 t-test; p = 0.006 KS2 test; n = 8 cells from six animals). To determine if this ratio was affected following prolonged activity, we also measured the five msec IPI paired pulse ratio >2 min after a tetanic stimulation (100 Hz for 4 s) and found that the paired pulse ratio was similar following tetanic stimulation (*Figure 6A₂*). In a separate set of experiments, we tested a different protein synthesis inhibitor, emetine (20 µM), for 1- to 2 hr to further verify the effects of inhibiting protein synthesis (*Maharana et al., 2013*; *Baleriola et al., 2014*). Similar to treatment with anisomycin, the average peak amplitude for emetine treated neurons (6.19 ± 0.9 nA) was smaller than the peak amplitude of control neurons (7.95 ± 1.0 nA) at the onset of 100 Hz stimulation (p = 0.1 t-test; p = 0.078 Wilcoxon Signed Rank test). Emetine treatment also produced an increase in paired pulse responses (*Figure 6—figure supplement 1*). At an interval of 5 msec, the paired pulse ratio increased from 0.56 ± 0.1 SEM in control recordings to 0.78 ± 0.1 SEM after treatment with emetine (*Figure 6—figure supplement 1A,B*; p = 0.03, paired t-test; p = 0.03 WSR test; emetine AD-test for normal distribution p = 0.03; seven control recordings paired with seven emetine recordings; seven mice total). In summary, given the results from two different protein synthesis inhibitors, we conclude that ongoing protein synthesis affects vesicle release as demonstrated by the changes in the paired pulse ratio.

Next, we compared EPSCs in control and protein synthesis inhibited conditions throughout a 400 msec train at 200 Hz stimulation (*Figure 6B₁*). Consistent with the paired pulse results, we observe a reduced level of depression throughout the train of EPSCs after inhibiting protein synthesis with anisomycin compared to the response from control neurons. Accordingly, we directly compared the average responses for the entire train and found that the reduced depression seen after inhibiting protein synthesis occurred throughout the 400 msec train at 200 Hz (*Figure 6B₂*). In contrast, for blinded experiments in which DMSO was applied, the treated and untreated responses were unchanged (*Figure 6—figure supplement 2A*). Finally, in a separate set of experiments, using emetine to inhibit translation we also find a reduction in depression during stimulation at 200 Hz (*Figure 6—figure supplement 3A*). Therefore, inhibiting protein synthesis results in higher relative levels of neurotransmitter release, which can be sustained (up to 80 EPSCs in *Figure 6B₂*) even at a brief five msec IPI.

Facilitation and depression are affected by the interval between pulses. Accordingly, we also measured the paired pulse ratio at 10 msec IPI (*Figure 6C1and C2*). At this interval, additional depression occurred in control recordings (0.50 ± 0.06 SEM, n = 8 cells from seven animals) but protein synthesis inhibited responses exhibited only weak depression (0.93 ± 0.06 SEM, n = 8 cells from seven animals; p < 0.001). We also tested how prior exposure to prolonged stimulation (4 s at 100 Hz) affects the paired pulse ratio and find a small change in control neurons (0.60 ± 0.07 SEM, n = 8 cells), but no change in the protein synthesis inhibited responses (0.95 ± 0.11 SEM, n = 8 cells). This

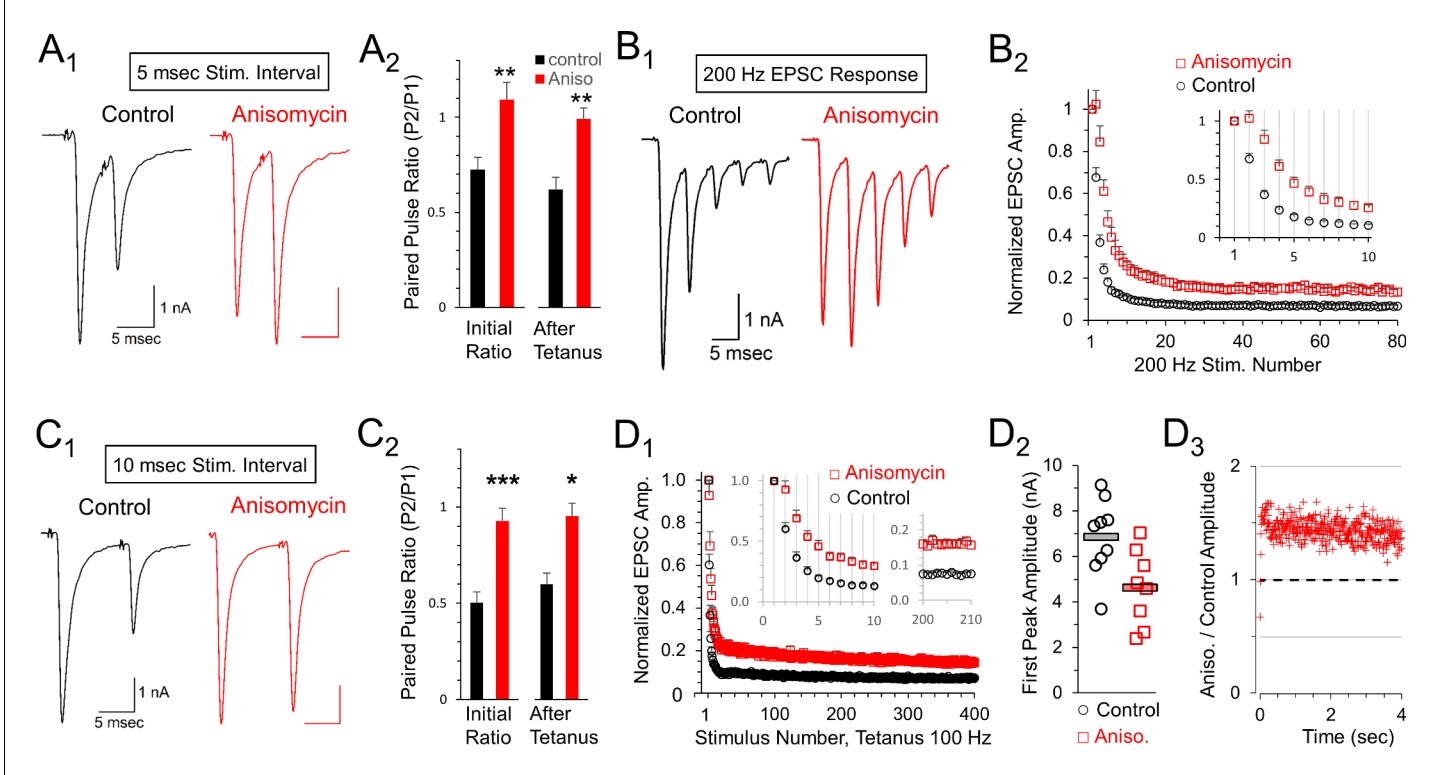

**Figure 6.** Reduced depression at 100 or 200 Hz after protein synthesis inhibition. ($A_1$) Representative traces of initial paired pulse responses, 5 msec interpulse interval (IPI). ($A_2$) Initial paired pulse ratio (5 msec IPI) for control (black, n = 9) and anisomycin (red, n = 8) treatment (p = 0.004 t-test, 0.006 KS2-test), compared to the ratio after 100 Hz tetanic stimulation for 4 s (p = 0.001 t-test; p = 0.006 KS2-test; n = 7 control, six anisomycin). Average Rs values for control and anisomycin were ~4.8 MΩ and ~7.6 MΩ, respectively. ($B_1$) Representative EPSC responses at 200 Hz stimulation for control and anisomycin. ($B_2$) Average normalized EPSCs at 200 Hz stimulation for control and anisomycin treatment (p < 0.001 ANOVA). Average Rs values for control and anisomycin were ~4.8 MΩ and ~7.6 MΩ, respectively. ($C_1$) Representative initial paired pulse responses at a 10 msec interpulse interval (IPI). ($C_2$) Initial paired pulse ratio (10 msec IPI) for control (black, n = 8) and anisomycin (red, n = 8) treatment (p < 0.001 t-test; p = 0.0014 KS2-test), compared to the ratio (10 msec IPI) after tetanic stimulation at 100 Hz for 4 s (p < 0.016 t-test; p = 0.049 KS2-test). ($D_1$) Average normalized EPSCs from 100 Hz tetanic stimulation in control and anisomycin treated neurons (p < 0.001 ANOVA). ($D_2$) Amplitudes of first EPSC in tetanic stimulation response for each control neuron (black circles) and anisomycin (red squares) treated neurons (p < 0.015 t-test; p = 0.034 KS2-test). Horizontal bars correspond to the average responses. ($D_3$) Ratio of 100 Hz EPSC responses from anisomycin and control neurons shows anisomycin treated neurons maintain a higher EPSC responses during sustained stimulation for 4 s.

DOI: https://doi.org/10.7554/eLife.36697.013

The following source data and figure supplements are available for figure 6:

**Source data 1.** Paired pulse ratios and peak responses at 100 Hz and 200 Hz following anisomycin treatment.
DOI: https://doi.org/10.7554/eLife.36697.020

**Figure supplement 1.** Paired pulse ratios (PPR) in control and after protein synthesis inhibition induced by emetine treatment.
DOI: https://doi.org/10.7554/eLife.36697.014

**Figure supplement 1—source data 1.** Emetine paired pulse ratios.
DOI: https://doi.org/10.7554/eLife.36697.015

**Figure supplement 2.** Peak amplitudes for blinded experiments for cells that were randomly treated with DMSO instead of anisomycin.
DOI: https://doi.org/10.7554/eLife.36697.016

**Figure supplement 2—source data 1.** DMSO peak amplitudes at 100 Hz and 200 Hz stimulation.
DOI: https://doi.org/10.7554/eLife.36697.017

**Figure supplement 3.** Normalized average excitatory postsynaptic currents in control neurons and neurons treated with emetine to inhibit protein synthesis.
DOI: https://doi.org/10.7554/eLife.36697.018

**Figure supplement 4.** Amplitude and series resistance values during tetanus recordings.
DOI: https://doi.org/10.7554/eLife.36697.019

indicates that the lack of paired pulse depression in protein synthesis inhibited synaptic responses is stable. In a separate set of experiments, we also tested the effects of inhibiting protein synthesis with emetine. At an interval of 10 msec, the paired pulse ratio increased from $0.55 \pm 0.08$ SEM in control recordings to $0.80 \pm 0.1$ SEM after inhibiting protein synthesis with emetine (*Figure 6—figure supplement 1C,D*; p = 0.01 paired t-test; 0.015 WSR-test; emetine data AD-test for normal distribution = 0.084; n = 7 control and seven emetine recordings; seven mice total). In summary, the differences we observe in the paired pulse depression at 5 and 10 msec IPI are consistent with differences in presynaptic mechanisms involving vesicle release (*von Gersdorff and Borst, 2002*; *Fioravante and Regehr, 2011*), indicating that inhibiting translation has an effect on vesicle release resulting in reduced paired pulse depression.

To determine how the paired pulse depression at a 10 msec IPI affects responses throughout prolonged trains, we measured all EPSCs in response to a 4 s tetanic stimulation at a 10 msec IPI (100 Hz). Similar to the results at a five msec IPI (200 Hz), we find inhibiting protein synthesis by treatment with anisomycin reduced depression at 100 Hz compared to control responses (*Figure 6D$_1$*). This effect was not seen when DMSO was applied during the blinded experiments (*Figure 6—figure supplement 2B*). In separate experiments using emetine to inhibit translation, we also find a reduced amount of depression for responses at 100 Hz (*Figure 6—figure supplement 3B*). Therefore, inhibiting protein synthesis with either anisomycin or emetine reduces depression during both 100 Hz and 200 Hz stimulation.

Facilitation or reduced paired pulse depression often occurs when the initial probability of release is reduced (*Fioravante and Regehr, 2011*), although other mechanisms may exist (*Neher, 2017*). In agreement with this, we note that the average P1 amplitude is smaller for anisomycin treated neurons ($4.61 \pm 0.59$ SEM) compared to controls ($6.85 \pm 0.59$ SEM; p = 0.015; *Figure 6D$_2$*). To further measure differences in the responses, we also graphed the non-normalized peak responses in protein synthesis inhibited conditions divided by control conditions (*Figure 6D$_3$*). Aside from the first three responses, the average peak amplitudes of the protein synthesis inhibited responses were ~1.4 fold higher than the peak amplitudes of control responses. This demonstrates that the translationally inhibited cells are maintaining higher amounts of release, due to lower depression, compared to responses from control cells even during prolonged stimulation for 4 s (*Figure 6D$_3$*) indicating that this resistance to depression is robust. We note that it is completely possible that the effects of inhibiting protein synthesis could vary with the speed and extent of inhibition, and could also change with repeated activity occurring over several hours, or be accompanied by other changes in synaptic responses. Therefore, enhanced synaptic response may be an initial consequence of inhibiting protein synthesis, and different or additional effects are likely to occur over hours or days.

## Inhibiting protein synthesis affects vesicle release and replenishment

The presynaptic effects on paired pulse ratios and increased response levels that occur throughout trains of prolonged stimulation suggest that the readily release pool (RRP), vesicle release, and the vesicle replenishment rate could be affected by inhibiting protein synthesis. To measure this, we graphed the cumulative EPSC response during a 100 Hz stimulation train (*Figure 7A$_1$*). A best fit line through the EPSC responses to stimuli 20 to 30 (*Figure 7A$_1$*) provided the slope of the steady state response, and the y-intercept of this line provides a measurement of the readily releasable pool (*Schneggenburger et al., 1999*; *Neher, 2015*). Using this method, the average size of the RRP (*Figure 7A$_2$*) was the same for control neurons ($16.3 \pm 1.6$ nA, n = 9 control and eight anisomycin cells) and protein synthesis inhibited neurons (anisomycin, $15.97 \pm 1.9$ nA, n = 8 from seven animals; p = 0.89 t-test; 0.88 KS2-test). Therefore, the initial capacity for vesicular release of neurotransmitter is not changed by inhibiting protein synthesis. Next, we find the initial release probability (Pr) for control neurons ($0.44 \pm 0.03$) is higher than the value in anisomycin treated neurons ($0.29 \pm 0.02$; p = 0.003 t-test; p = 0.006 KS2-test), therefore the differences we see in the initial peak response in protein synthesis inhibited neurons is explained by a reduction in the initial probability of release (*Figure 7A$_3$*). Lastly, the slope of the steady state of the cumulative response (*Figure 7A$_1$*) provides a measurement of the rate of vesicle replenishment. We find that the rate of vesicle replenishment (*Figure 7A$_4$*) increases when protein synthesis is inhibited ($96.9 \pm 1.6$ pA/msec), compared to the rate in control neurons ($62.2 \pm 3.5$ pA/msec; p = 0.037 t-test; p = 0.041 KS2-test; control data AD-

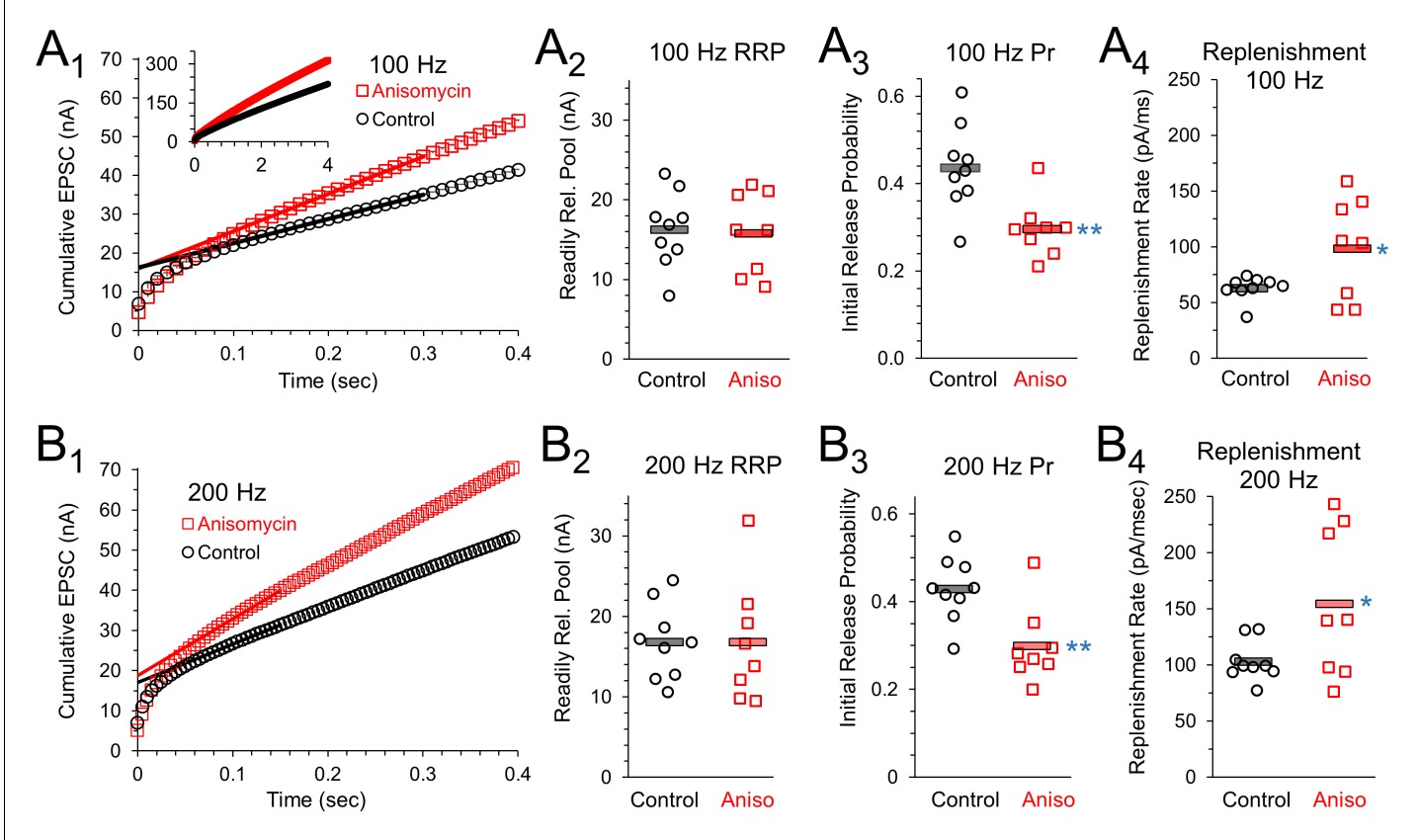

**Figure 7.** Initial release probability and vesicle release during sustained activity are affected by inhibiting protein synthesis. (A₁) 100 Hz Cumulative EPSC plot of averaged peak response in control (black) and protein synthesis inhibited (anisomycin, red) neurons. The y-intercept of a line fit to responses 20 to 30 measures the readily releasable pool (RRP). (A₂) RRP measurement for each cell, during 100 Hz stim, measured as the y-intercept of a best fit line through responses 20 to 30, for the beginning of the steady state response. (A₃) The initial probability of release (Pr₁) for 100 Hz trains, measured as the ratio of the amplitude of the first EPSC to the RRP. Averages shown by horizontal bars. (A₄) Rate of vesicle replenishment during steady state response during 100 Hz stimulation, measured as the slope of a best fit line through responses 20 to 30, separately measured for each cell. (B₁) 200 Hz Cumulative EPSC plot of averaged peak response in control and protein synthesis inhibited (anisomycin) neurons. The y-intercept of a line fit to responses 20 to 30 measures the readily releasable pool (RRP). (B₂) RRP measurement for each cell, during 200 Hz stim, measured as the y-intercept of a best fit line through responses 20 to 30, for the beginning of the steady state response. (B₃) The initial probability of release (Pr₁) measured during 200 Hz trains, measured as the ratio of the amplitude of the first EPSC to the RRP. Averages shown by horizontal bars. (B₄) Rate of vesicle replenishment during the steady state response during 200 Hz stimulation, measured as the slope of a best fit line through responses 20 to 30, separately measured for each cell.

DOI: https://doi.org/10.7554/eLife.36697.021

The following source data and figure supplements are available for figure 7:

**Source data 1.** Vesicle replenishment, initial release probability, and readily releasable pool following anisomycin treatment.
DOI: https://doi.org/10.7554/eLife.36697.024

**Figure supplement 1.** Vesicle release properties after inhibiting protein synthesis by treatment with emetine.
DOI: https://doi.org/10.7554/eLife.36697.022

**Figure supplement 1—source data 1.** Relative replenishment, initial release probability, and readily releasable pool following emetine treatment .
DOI: https://doi.org/10.7554/eLife.36697.023

test for normal distribution p = 0.015). This indicates that an increased rate of vesicle replenishment is responsible for the increased responses following anisomycin treatment (*Figure 6B2, D1and D3*).

To provide additional estimates of vesicle release properties, we also measured responses from stimulation at 200 Hz (*Figure 7B₁*). As anticipated, we found that the readily releasable pool values for control and protein synthesis inhibited neurons are the same at 100 Hz and 200 Hz (*Figure 7A₂ and B₂*). At 200 Hz, the RRP-value for control neurons is $16.8 \pm 1.5$ nA, which matches the RRP-value for anisomycin treated neurons ($16.8 \pm 2.6$ nA; $p > 0.99$ t-test; 0.88 KS2-test). As expected, the initial

probability of release at 200 Hz is nearly identical to the values measured at 100 Hz stimulation (*Figure 7A₃*), with a Pr of 0.43 ± 0.02 for control and 0.3 ± 0.03 for anisomycin treated neurons (p = 0.005 t-test; p = 0.006 KS2-test; anisomycin AD-test for normal distribution p = 0.08) estimated from the EPSC responses generated by 200 Hz stimulation (*Figure 7 B₃*). However, the rate of vesicle replenishment, as measured by the slope of the steady state response in the cumulative EPSC graph, is higher at 200 Hz compared to 100 Hz (*Figure 7A₄ and B₄*), which is expected given the need to maintain release levels. At 200 Hz, the anisomycin treated neurons maintained a faster rate of replenishment (154.1 ± 23.4 pA/msec) compared to control neurons at 200 Hz (102.8 ± 5.9 pA/ msec; p = 0.04 t-test and KS2-test; control data AD-test for normal distribution p = 0.053). We note that the rate of vesicle replenishment in control neurons at 200 Hz (102.8 ± 5.9 pA/msec) was similar to the rate of vesicle replenishment in anisomycin treated neurons at 100 Hz (96.9 ± 1.6 pA/msec; p = 0.72 t-test, 0.49 KS2-test). Interestingly, the magnitude of the increase in the rate of vesicle replenishment from 100 to 200 Hz stimulation is nearly identical for control (1.65) and anisomycin treated neurons (1.59). The finding that the increase in the vesicle replenishment rate that occurs from 100 to 200 Hz is nearly the same in control and anisomycin treated neurons indicates that the factors that scale up the rate of replenishment are not affected by inhibiting protein synthesis. Instead, it appears that the normal steady state levels of vesicle release and vesicle replenishment are increased after protein synthesis is inhibited.

In addition to the anisomycin experiments, we also tested vesicle release properties after inhibiting protein synthesis with emetine at 20 µM for 1–2 hr. Due to a range of peak amplitudes on different recording days, we normalized cumulative release for control and emetine treated conditions and we find a steeper slope for recordings from cells treated with emetine, compared to control (*Figure 7—figure supplement 1A,D*). The slope of the steady state portion of this graph, corresponding to the relative rate of replenishment, is 18.5 for emetine treated neurons and 15.1 in the controls at 100 Hz stimulation (*Figure 7—figure supplement 1A*). Similarly, at 200 Hz emetine treated neurons have a faster relative rate of replenishment, which is 24.9 compared to 20.4 in control recordings (*Figure 7—figure supplement 1D*). We also find a decrease in the probability of release (Pr) following treatment with emetine. At 100 Hz, Pr decreases from 0.46 ± 0.07 SEM in controls to 0.33 ± 0.06 SEM (p = 0.036 paired t-test; 0.047 Wilcoxon Signed Rank-test; emetine AD-test for normal distribution p = 0.01; n = 7 for each condition; seven mice) after inhibiting protein synthesis with emetine (*Figure 7—figure supplement 1B*). A similar reduction in Pr is seen with 200 Hz stimulation, where the Pr under control conditions was 0.47 ± 0.06 SEM, and decreased to 0.36 ± 0.05 SEM (p = 0.046 paired t-test; p = 0.031 WSR-test; emetine AD-test for normal distribution p = 0.01; n = 7 for each condition; seven mice) after inhibiting protein synthesis with emetine (*Figure 7—figure supplement 1E*). In contrast, the RRP was unchanged at 100 Hz (control 19.2 ± 1.1 nA; emetine = 20.9 ± 1.3 nA; p = 0.38 paired t-test; p = 0.58 WSR-test; n = 7) and 200 Hz (control = 17.8 ± 2.1; emetine = 19.5 ± 3.2; p = 0.38 paired t-test; p = 0.58 WSR-test; n = 7). To provide an additional measurement for the Pr and the RRP, we fit exponentials to the EPSC decay during stimulation (*Figure 7—figure supplement 1C,F*) and calculated the Pr and RRP as described in the Materials and methods section. Using this method, we find a similar trend for a decrease in the Pr after inhibition with emetine, and we find no major difference in the RRP. A fit to the average data at 100 Hz gives a Pr of 0.53 in control neurons which decreased to a Pr of 0.36 after emetine treatment. At 200 Hz, in control neurons, the Pr was 0.48, compared to a Pr of 0.35 after treatment with emetine. However, the RRP size was similar in control neurons (15.1 nA) and in emetine treated neurons (17.6 nA) at 100 Hz stimulation. Similarly, at 200 Hz stimulation, the RRP was 17.2 nA in control recordings and 19.5 nA after inhibiting translation with emetine. Therefore, using two different drugs, we find a reduction in the initial Pr, an increase in the rate of vesicle replenishment, and a similar size of the RRP in control cells and translation inhibited cells.

## Discussion

We hypothesized that local translation occurs in the presynaptic compartment and that it is necessary to maintain normal levels of neurotransmitter release. In support of this, we have shown that 5.8S rRNA, a major component of ribosomes, is present in the presynaptic terminal at the calyx of Held synapse. This provides further evidence that presynaptic ribosomal components are present in established mammalian CNS nerve terminals (*Younts et al., 2016*) in addition to developing neurites

and axons (*Taylor et al., 2013*; *Batista et al., 2017*). We verified that presynaptic ribosomes are functional, using the SUnSET technique which has previously been used to demonstrate local translation in dendritic, axonal and neuritic compartments (*Batista et al., 2017*) in cell cultures. Due to the large size of the calyx of Held we were able to employ this technique in mammalian brain slice to show that presynaptic ribosomes are present and functional, producing a presynaptic signal within tens of minutes. Notably, this signal showing active translation at functional ribosomes is completely blocked when brain slices are pretreated with a translational inhibitor. These data provide convincing evidence that local protein synthesis occurs at the calyx of Held nerve terminal.

In addition to demonstrating active translation in presynaptic ribosomes in the calyx of Held, we find that ongoing protein synthesis affects neurotransmitter release. Starting with spontaneous activity, we found that inhibiting translation causes an increase in the frequency of spontaneous release. Specifically, the spontaneous release events show a population of spontaneous release events with brief inter-event intervals, and a population that has longer intervals between events. We find that inhibiting translation effectively increases the frequency of spontaneous release by increasing the percentage of brief event intervals to ~65% of the spontaneous release intervals. In control neurons we found the opposite relationship, where longer event intervals account for ~65% of spontaneous release events. Therefore, ongoing protein synthesis limits spontaneous release by favoring longer inter-event intervals. The finding that ongoing protein synthesis acts to lower the spontaneous release frequency is consistent with the idea that the frequency of spontaneous release is highly controlled, given that spontaneous events function as important signals in synaptic development and homeostasis (*Kavalali, 2015*). For example, spontaneous release of glutamate can suppress local protein synthesis in dendrites (*Sutton et al., 2006*). Furthermore, NMDA receptors that are specifically activated by spontaneous release appear to be responsible for the rapid antidepressant effect produced by ketamine exposure (*Autry et al., 2011*). Therefore, over the last decade, accumulating evidence demonstrates that spontaneous release can have significant effects on synaptic function. In addition, vesicles that undergo spontaneous release may preferentially come from a population separate from the vesicles that respond to evoked release (*Sara et al., 2005*), and this appears to involve association with specific proteins (*Hua et al., 2011*). Finally, vesicles involved in spontaneous release can have different sensitivity to intracellular calcium than vesicles involved in evoked release, and spontaneous release may be able to occur independent of intracellular calcium (*Schneggenburger and Rosenmund, 2015*). Therefore, vesicles that undergo spontaneous release appear to be controlled separately from vesicles that fuse in response to action potentials. Spontaneous release is a highly regulated process that is important in maintaining synaptic function, and we show that ongoing protein synthesis plays a role in limiting the frequency of spontaneous release events.

Tetanic stimulation has previously been shown to transiently increase the frequency of spontaneous release, due to residual calcium (*Korogod et al., 2005*) and may also involve PKC activation (*Korogod et al., 2007*; *Fioravante et al., 2011*). In our work, we show an increased frequency of spontaneous release following tetanic stimulation. Despite having initially opposite levels of fast and slow components of spontaneous release in the graph of the cumulative probability of event intervals, control and protein synthesis inhibited responses have matching percentages of fast and slow components following a tetanic stimulation. Accordingly, the increased spontaneous release that occurs after inhibiting protein synthesis may in part involve similar mechanisms that increase spontaneous release after tetanic stimulation. Alternatively, other factors have been shown to affect the rate of spontaneous release. For example, inhibiting myosin light chain kinase (MLCK) at the calyx of Held causes an increase in the rate of spontaneous release (*Srinivasan et al., 2008*).

In addition to the effects on spontaneous release, we also observe facilitation or reduced depression in paired pulse ratios when protein synthesis is inhibited. Interestingly, although we found that the readily releasable pools are identical in protein synthesis inhibited and control conditions, the initial probability of release during a train of stimulation is lower when protein synthesis is inhibited. However, during prolonged stimulation, a reduced amount of depression is maintained throughout the stimulation. It has been shown that presynaptic release probability can be modulated by a small group of presynaptic proteins. shRNA knockdown experiments of a vertebrate-specific protein, Mover, in the calyx of Held nerve terminal results in an increase in the probability of release. This demonstrates that expression of specific presynaptic proteins can negatively or positively influence synaptic release probability (*Körber et al., 2015*). Interestingly, in addition to affecting spontaneous

release, MLCK inhibition has also been shown to cause an initial increase in the amplitude of EPSCs during high frequency stimulation, although this appears to involve changes in the size of the readily releasable pool (*Srinivasan et al., 2008*). In our work, we find an increase in the rate of vesicle replenishment when protein synthesis is inhibited, indicating that the higher levels of release require a faster rate of vesicle replenishment to maintain the elevated amount of release (*Sara et al., 2002*; *Qiu et al., 2015*). Therefore, despite the reduced probability of release for the first response in the stimulation trains, the overall levels of vesicle release and subsequent levels of vesicle replenishment are elevated after inhibiting protein synthesis.

Importantly, the shape and amplitude of spontaneous and evoked responses are unaffected during the 1–2 hr that protein synthesis is inhibited in our experiments, indicating that the postsynaptic receptor responses are not affected. The lack of change in the postsynaptic membrane currents also argues against a possible retrograde signal that is affected by inhibiting protein synthesis. In support of this, in experiments where the postsynaptic neuron was lysed and removed, presynaptic release properties were not changed (*He et al., 2006*). Therefore, the differences in synaptic responses under control and protein synthesis inhibited conditions are due to presynaptic effects on neurotransmitter release. The effect on spontaneous and evoked neurotransmitter release properties, combined with the presence of active presynaptic ribosomes, indicate that presynaptic proteins which act to limit synaptic transmission can be synthesized locally in the presynaptic terminal.

There is evidence that a majority of synaptic proteins have a half-life of ~36 hr (*Cohen and Ziv, 2017*) however the techniques used to measure turnover may not be sensitive enough to capture proteins with a faster turnover rate that could compose a small but functionally important population of presynaptic proteins. In addition, it would not be surprising if neuronal stimulation induced the turnover of synaptic proteins, independent of their normal turnover rate (*Alvarez-Castelao and Schuman, 2015*). Whether degradation, via the proteasome pathway, itself is regulated by activity still remains an intriguing question. For example, it has been shown that treatment of neurons with activity blockers results in a decrease of ~50% in the polyubiquitinated profile of proteins localized in post-synaptic density (PSD) fractions, whereas treatment of neurons with activity inducers resulted in an increase in polyubiquitinated proteins (*Ehlers, 2003*). This would facilitate the need for rapid on-site protein production. More extensive studies identifying proteasome substrates in the context of neuronal activity would contribute to a better understanding of the role of protein turn over in synaptic plasticity.

The ability to synthesize some proteins locally makes particular sense in cells that have long processes, such as axons and dendrites. The transient rate of axoplasmic transport has been reported to ~1 µm/sec in calyx of Held axons (*Wimmer et al., 2004*), and sustained rates are much slower (*Maday et al., 2014*). In addition, translation at the cell body requires retrograde axonal transport of a signal from the nerve terminal to the cell body, followed by subsequent production of protein and the anterograde axonal transport of protein for delivery to the nerve terminal. Therefore, local synthesis of at least some regulatory proteins saves significant time, allowing a local neuronal region to rapidly upregulate some essential proteins in response to changes in neuronal activity. While significant work has been done over the last two decades to demonstrate the presence and properties of local protein synthesis in dendritic compartments of CNS neurons (*Rodriguez et al., 2008*; *Rangaraju et al., 2017*), evidence demonstrating the presence and requirements for local presynaptic protein synthesis in intact CNS mammalian neurons is very recent (*Younts et al., 2016*). However, work using mammalian synaptosomes has produced evidence of mRNA transcripts and the ability to generate newly synthesized proteins (*Alvarez et al., 2000*), although potential contamination from postsynaptic neurons or glia has been a major concern. Additional evidence comes from mRNA found in axons (*Alvarez et al., 2000*) and recently formed nerve terminals (*Batista et al., 2017*). These presynaptic transcripts code for a variety of proteins including some that can affect vesicle replenishment and fusion such as β-catenin, β-tubulin, and β-actin. In addition, transcripts for nuclear encoded mitochondrial proteins have been found in axons. It is important to note that in our experiments, the calyx nerve terminal is no longer connected to the neuronal cell body because the axons are severed during brain slicing. Given the lack of connection between the cell body and nerve terminal, newly synthesized proteins in the calyx nerve terminal cannot come from the cell body that gives rise to the axon that forms the nerve terminal. Based on our imaging data, we conclude that ongoing protein synthesis is occurring in the presynaptic terminal, although some amount could also occur in the adjacent section of the axon.

The work shown here is the first to directly show local protein synthesis occurring in established nerve terminals, in situ, in mammalian brain slices. This agrees well with earlier findings that ribosomes are present in presynaptic terminals in hippocampal interneurons in brain slice (*Younts et al., 2016*). In addition, this group clearly demonstrated that presynaptic protein synthesis is necessary for a form of long-term depression of inhibitory transmission. Our finding that ongoing protein synthesis can serve to limit vesicle release (*Figure 6 B$_2$, D$_1$, D$_3$* and *Figure 7 A$_1$, A$_4$, B$_1$, B$_4$*) has interesting implications for how nerve terminals maintain and modulate their presynaptic release properties. Limiting evoked synaptic responses effectively increases presynaptic efficiency by allowing the nerve terminal to conserve some of the substantial energy involved in vesicular release, retrieval, refilling, and replenishment of readily releasable vesicles (*Rangaraju et al., 2014*; *Shulman et al., 2015*; *Sobieski et al., 2017*). Limiting synaptic release should also help the nerve terminal to maintain sufficient responses for a longer time. This indicates an important function for newly synthesized proteins over a time course that doesn't allow transport from the cell body to the presynaptic terminal. The further study of the presynaptic processes that are affected by inhibiting translation will help us to better understand how spontaneous and evoked release are controlled, and the role that local protein synthesis plays in maintaining and modulating synaptic responses.

## Materials and methods

### Slice preparation and electrophysiology

Brain slices

C57BL6 mice (Charles River Laboratories) from postnatal day 8 to 12, of either sex were used for all experiments described. The mice were housed in a facility approved by the Association for Assessment and Accreditation of Laboratory Animal Care International, and protocols used for handling and care were reviewed by the Rutgers University Animal Care and Facilities Committee. Animals were decapitated without prior anesthesia, in accordance with NIH guidelines. Transverse brainstem slice thickness varied from 100 µm (immunohistochemistry and imaging) to 180 µm (electrophysiology) and were generated using a Leica VT1200 vibratome. Throughout the process of dissection and slicing, the brain was maintained in a low-calcium artificial CSF (aCSF) solution at 1–2°C containing the following (in mM): 125 NaCl, 25 NaHCO$_3$, 2.5 KCl, 1.25 NaH$_2$PO$_4$, 25 glucose, 0.8 ascorbic acid, three myo-inositol, 2 Na-pyruvate, 3MgCl$_2$, and 0.1 CaCl$_2$, pH 7.4, when oxygenated with carbogen gas (95% oxygen, 5% carbon dioxide). Once produced, slices were transferred to a holding chamber maintained at ~35°C for 30–40 min in normal calcium aCSF solution with the same composition listed above except for 1 mM MgCl$_2$ and 2 mM CaCl$_2$. This same solution was also used as the standard recording solution for electrophysiology experiments (see below). All experiments were performed at room temperature (22–25°C) for up to ~4–5 hr after the recovery period.

Electrophysiology

Patch-clamp recordings were conducted using an EPC10 USB double patch-clamp amplifier with PatchMaster software (HEKA; Harvard Bioscience). A transverse slice orientation was used in all postsynaptic voltage-clamp recordings in order to maintain the integrity of the calyceal axons for fiber stimulation. Calyx synapses in the medial nucleus of the trapezoid body (MNTB) were afferently stimulated (A-M Systems Isolated Pulse Stimulator Model 2100) using a bipolar fiber stimulator (lab design) placed at the midline of the slice. The MNTB field was scanned with an extracellular pipette to locate neurons that respond to midline fiber stimulation. For whole-cell recording, patch pipettes were produced from thick-walled borosilicate glass, 2.0 mm outer diameter, 1.16 mm inner diameter (Sutter Instruments). Postsynaptic pipettes (2–3 MΩ) were filled with a solution containing (in mM): 125 Cs-methanesulfonate, 20 CsCl, 20 TEA, 10 HEPES, five phosphocreatine (Alpha Aesar), 4 ATP, 0.3 GTP, and 2 QX-314 Cl$^-$ (Sigma Aldrich; to block voltage gated Na$^+$ channels on the postsynaptic neuron to measure the true EPSC) and was buffered to pH 7.4 using CsOH. To inhibit protein synthesis, anisomycin (Sigma, A9789) and emetine dihydrochloride (EMD Millipore, Calbiochem, 324693). Postsynaptic series resistances (R$_s$) for voltage clamp recordings was less than 15 MΩ and typically varied less than 2 MΩ throughout the recording. For the recordings that appear in *Figures 5*, *6* and *7*, showing EPSC response properties in control and after anisomycin treatment, the average R$_s$ values and corresponding peak EPSC amplitudes in response to stimulation at 100 Hz

and 200 Hz are graphed (see *Figure 6—figure supplement 4A,B*). We also provide a graph of Rs values and corresponding EPSC peak amplitude for the data in *Figures 6* and *7* (see *Figure 6—figure supplement 4C,D,E,F*). In addition, an $R_s$ compensation of 75–80% was applied for all recordings such that the adjusted $R_s$ was in the range of 2–5 MΩ. Cells for which these criteria could not be applied, or maintained, were excluded from analysis. Recordings were acquired at sampling frequencies of 20 KHz and filtered by a 4-pole Bessel filter at 3 kHz. Holding potentials were set to −65 mV; junction potentials, calculated to be −11 mV, were not corrected.

## Blinded testing and analysis conditions and criteria for testing anisomycin

For all electrophysiology recordings and data analysis measurements, anisomycin treatment and control conditions were blinded. In addition, the quality of the slices, neurons, and general recording conditions were determined by 1–2 initial recordings in normal aCSF. If the initial recordings had stable responses that lasted the duration of the stimulus protocols, a minimum of 25 min, then the recording solution was switched to a blinded cylinder of recording solution for subsequent recordings which were performed after ~1 hr of treatment (45 min to 120 min) in the absence of fiber stimulation. Since spontaneous action potentials are not present in these recording conditions, only spontaneous release activity occurred during the treatment period. On each day, the blinded cylinder would contain either: 40 µM anisomycin (Sigma Aldrich) or DMSO alone (vehicle). To test the effect of the translational inhibitor anisomycin (40 µM) on synaptic response characteristics, slices were preincubated for ~1 hr in the presence of the drug in the absence of afferent fiber stimulation. At all times, aCSF was continuously circulated using a peristaltic pump; total volume of the solution was 30 mL. All recordings, control and test conditions, were made in the presence of 25 µm bicuculline and 2 µm strychnine to block inhibitory responses. Power analysis to determine the appropriate sample size was performed based on means and standard deviation values of preliminary data. Recordings from 5 cells in each condition was estimated to be adequately powered, for α = 0.5, and a 0.8 power of test.

## Recordings from MNTB neurons and Data Analysis

Miniature excitatory post-synaptic currents (mEPSCs) were recorded during 30 s continuous recordings at several times during the stimulation protocol. mEPSCs were analyzed by Mini Analysis Software (Synaptosoft, RRID:SCR_002184). The following mEPSC search parameters were used: gain, 20; blocks, 3940; threshold, 10 pA; period to search for a local maximum, 20,000 µsec; time before a peak for baseline, 5000 µsec; period to search a decay time, 5000; fraction of peak to find a decay time, 0.5; period to average a baseline, 2000 µsec; area threshold, 10; number of points to average for peak, 3; direction of peak, negative). Analysis was performed using the above settings, and visually checked to ensure accuracy. Evoked response traces were exported to Igor Pro (Wavemetrics, Portland, OR), and measurements were made manually, or using Taro Tools (Igor macro, Taro Ishikawa) with visual inspection and adjustment as necessary for every measured peak amplitude.

Data are presented as mean ± standard error of the mean (SEM). A Student's t-test (MS Excel) was employed to determine if statistically significant differences exist between treated and control conditions. While the t-test is considered to be robust, data that do not have a normal distribution can affect the p-value. Therefore, an Anderson-Darling test (MATLAB) for normal distribution was run on each dataset used in the t-test comparisons. The null hypothesis for this test is that the dataset is normally distributed. Typically, α is set at 0.05, and we provide p-values for all Anderson-Darling test p-values < 0.1. Therefore, unless otherwise noted, p > 0.1 for each set of data. To further account for the possibility that the distribution of a dataset could affect the t-test value, the two-sample Kolmogorov-Smirnov (KS2) test (MATLAB), a nonparametric method, was also used to determine statistical significance. For the paired data comparisons, a two-sided Wilcoxon signed rank test (MATLAB) was used as a nonparametric test. For the repetitive stimulation data, a two way ANOVA (MS Excel) with repeated measures was used to compare the normal and treated responses. To reduce complications for running an unbalanced ANOVA, in a single data set (100 Hz anisomycin), the values from two recordings done on one day were averaged to allow an equal number of recordings to run a balanced two way ANOVA. Calculated *p*-values are indicated in relevant figures as follows: $p \leq 0.05$ is considered significant (*); $p \leq 0.01$ very significant (**); and $p \leq 0.001$ highly significant (***). We define biological replicates as each tested cell (number of recordings), and

technical replicates as multiple tests on a single cell. In our experiments, a minimum of four recordings, of spontaneous activity, 30 s each, were made during the recording time. Data were analyzed as initial spontaneous release levels, and spontaneous release following activity as described in the text. Outlier data for spontaneous event recordings resulted in removal of two cells from the data, as determined by Grubb's test with $\alpha$ = 1%. The two-sample Kolmogorov-Smirnov (KS2) test was used to calculate the $p$-value for the cumulative probabilities of the mEPSC event intervals for two different conditions. Briefly, this nonparametric test uses the maximum vertical difference between two cumulative probability graphs and the total number of measurements to determine the statistical significance of the differences between two cumulative probability distributions. Histograms with identical bin-ranges were used to compare the mEPSC intervals for the two different conditions. This calculation was preformed manually, and by the KS function in MATLAB, which gave very similar or identical values.

Vesicle release properties were measured by plotting the cumulative response for each recorded response to 100 and 200 Hz stimulation. A best fit line through the cumulative EPSC responses to stimuli 15 to 30 (*Figure 7A$_1$ and B$_1$*) provides the slope of the steady state response corresponding to the vesicle replenishment rate; and the y-intercept of this line provides a measurement of the readily releasable pool (*Schneggenburger et al., 1999*; *Neher, 2015*). The initial probability of release was measured by dividing the peak amplitude of the first EPSC response by the corresponding RRP measured for that train (*Schneggenburger et al., 1999*; *Neher, 2015*). As an additional method to measure the initial probability of release and the readily releasable pool, the peak response was plotted against the stimulation number and a single exponential was fitted to the data (*Thanawala and Regehr, 2016*). The tau ($\tau$) of the exponential fit was used to calculate the probability of release (Pr) using the formula: $Pr = (1-e-1/\tau)$. In instances where the response to the first stimulation ($R_0$) was smaller than the value predicted by the fitted exponential ($RP_0$), a facilitation correction (Fc) value was calculated by dividing the predicted value by the actual value: $Fc = RP_0 / R_0$, and Pr was divided by Fc to compensate for facilitation.

## Immunohistochemistry and confocal microscopy

### Immunohistochemistry

Either sex of C57BL6 mice, postnatal (PN) day 8 to 12, (n = 18) were decapitated without previous use of anesthesia, and transverse auditory brainstem slices (100–140 µm thick) were prepared as described above. Following recovery in normal aCSF, sections were transferred to a 12 well culture plate and washed 2x in phosphate buffered saline (PBS) (in mM: 137 NaCl, 2.7 KCl, 4.3 $Na_2HPO_4$*7$_H$20, and 1.4 $KH_2PO_4$, pH 7.4). Following the washes, the solution was replaced with ice-cold PBS containing 4% (wt/vol) PFA and fixed for 30 min at room temperature, with gentle agitation. After fixation the sections were rinsed 3x with PBS and incubated in blocking and permeabilization buffer in PBS containing 10% (vol/vol) normal goat serum (MP Biomedicals, LLC), 2% (wt/vol) BSA and 0.25% (vol/vol) Triton X-100 (Alfa Aesar) for 1h30m at room temperature. Slices were again rinsed with PBS 3x, 10 min for each wash. Sections were further blocked in PBS containing 40 µg/mL of AffiniPure Fab Fragment Goat Anti-Mouse IgG (H + L) for 1 hr at room temperature. Slices were then washed (3x) and placed in PBS containing the following; 1% (vol/vol) normal goat serum, 1% (vol/vol) BSA, 0.25% (vol/vol) Triton X-100, and mouse monoclonal anti-5.8S rRNA, clone Y10b at 1:500 (Abcam, ab37144, RRID: AB_777714) overnight at 4°C. Importantly, to minimize the possibility if non-specific interactions, all double labeling experiments were done sequentially. Following overnight incubation, slices were washed 3x in PBS, and placed in primary antibody solution containing guinea pig polyclonal anti-vesicular glutamate transporter 1 (VGLUT1) at 1:500 (Synaptic Systems, RRID: AB_887878) and allowed to incubate overnight at 4°C. Slices were rinsed 3x in PBS, and placed in PBS containing the following; 1% (wt/vol) BSA, 0.05% (vol/vol) Tween-20, and Alexa-594-conjuagted AffiniPure Goat Anti-Mouse IgG (H = L)(1:500) secondary antibody (Jackson, 115-585-003) for 2 hr at room temperature. Slices were rinsed 3x in PBS and then incubated in the same buffer as above, but with Alexa-488 conjugated AffiniPure Donkey Anti-Guinea Pig IgG (H + L) (1:500) secondary antibody (Jackson, 706-545-148) for 2 hr at room temperature. Sections were then washed with PBS and mounted to a glass slide, excess PBS was removed, a few drops of Fluoromount (Sigma Aldrich) was added and covered with Gold Seal cover slips #1.5 (Thermo Fisher). Slides were stored at 4°C.

To confirm 5.8S rRNA specificity we pretreated slices with nucleases. Following fixation sections were washed in PBS and incubated in PBS containing 0.25% (vol/vol) Triton –X100 for 45 min. Slices were then washed 3x (10 m each) in enzyme buffer (50 mM Tris and 5 mM $CaCl_2$, pH = 8). Next, slices were incubated in enzyme buffer containing 80 µg/mL RNase A (Fermentas, EN0531) and 300 U/mL micrococcal nuclease (New England Biolabs, M0247) for 60 m at 37°C. Note that control experiments were performed by incubating slices in enzyme buffer (with no enzymes) at 37°C. Following incubations, slices were rinsed 3x (10 m each) with PBS and blocked for 1h30m in PBS containing 10% (vol/vol) normal goat serum and 2% (wt/vol) BSA, at room temperature. Antibody application and further slice processing is the same as described above.

## Confocal Microscopy

Confocal image stacks of PFA treated brain slices were acquired using a Leica TCS SP2 laser-scanning microscope (Leica, Heidelberg, Germany). Image acquisition was performed in sequential scanning mode, using a 3-scan average for each image. Images were displayed and analyzed using FIJI. Regions of interest (ROI) were used to calculated image intensity of individual presynaptic terminals, postsynaptic principal cells, or both. Co-localization was performed by selecting neurons from the same-stacked images at random. Analysis was performed on the image where the neuron was widest. To reduce non-specific signal, a background subtraction value of 16 was subtracted from each image. The FIJI plugin JACop120 (just another co-localization plugin) was used for colocalization analysis which generated a graphical output table that contained the Pearson's correlation coefficient. Line scan analysis to qualitatively assess signal overlap in a given area was performed in FIJI using the line-plot profile feature. In all experiments performed the VGLUT1 signal was always higher in relative intensity than 5.8S rRNA and puromycin. The VGLUT1 signal marks the presence of synaptic glutamatergic vesicles which are specific to the presynaptic terminal. Therefore, regions that show colocalization with either the 5.8S rRNA or puromycin signal with the VGLUT1 signal demonstrates the presence of ribosomal components and ribosomal activity in the presynaptic compartment. We also note that the VGLUT1 signal does not label the entire presynaptic terminal. Therefore some regions that display a signal indicating ribosomal components or ribosomal activity may still be in the presynaptic compartment even though they do not have an overlapping VGLUT1 signal. This is particularly important where there appears to be a continuous presynaptic compartment that is only partially labeled by VGLUT1.

## SUnSET labeling of newly synthesized proteins

### Puromycylation Assay

Transverse brainstem slices were prepared as described above.

To detect newly synthesized proteins, slices were incubated in puromycin (1.8 µM Sigma Aldrich), added to normal aCSF, for 10 min following the brain slice recovery period. Control slices were pre-incubated first with 40 µM anisomycin for 60 min, and puromycin was added at the 50 min time point so that anisomycin and puromycin were present for the last ten minutes. Following incubation slices were rinsed 3x with pre-warmed PBS, prior to fixation. The slices were then further processed as described above, post-fixation. To detect puro-polypeptides a mouse monoclonal anti-puromycin antibody, clone 12D10 (EMD Millipore, MABE343; RRID: AB_2566826) was used at a dilution of 1:250. The secondary antibody used for indirect immunofluorescence was Alexa-594-conjuagted AffiniPure Goat Anti-Mouse IgG (H = L) at a dilution of 1:500. Imaging was carried out as described above.

## Acknowledgements

We thank Drs. Mark Plummer, Bonnie Firestein, Ulrich Hengst, Zhiping Pang, Daphné Robinson, Lu-Yang Wang, and Steve Clarke for discussions, critique, assistance, and suggestions on this work. We also thank Drs. Kelvin Kwan, Noriko Goldsmith, and Wise Young (W M Keck Center for Collaborative Neuroscience, Rutgers), Dr. Nanci Kane (Waksman Institute, Rutgers), Daniel Martin (Biomedical Engineering, Rutgers), Dr. Lorin Milescu and Dr. Mirela Milescu (University of Missouri) for advice, assistance, and use of imaging systems.

## Additional information

### Funding

| Funder | Grant reference number | Author |
| --- | --- | --- |
| National Institutes of Health | NS051401-42 | Kenneth G Paradiso |

The funders had no role in study design, data collection and interpretation, or the decision to submit the work for publication.

### Author contributions

Matthew S Scarnati, Formal analysis, Supervision, Validation, Investigation, Methodology, Writing—original draft, Project administration, Writing—review and editing; Rahul Kataria, Formal analysis, Investigation, Methodology, Writing—review and editing; Mohana Biswas, Formal analysis, Writing—review and editing; Kenneth G Paradiso, Conceptualization, Resources, Data curation, Formal analysis, Supervision, Funding acquisition, Validation, Investigation, Methodology, Writing—original draft, Project administration, Writing—review and editing

### Author ORCIDs

Matthew S Scarnati https://orcid.org/0000-0002-4306-7569
Kenneth G Paradiso http://orcid.org/0000-0002-6396-9412

### Ethics

Animal experimentation: This study was performed in strict accordance with the recommendations in the Guide for the Care and Use of Laboratory Animals of the National Institutes of Health. Every effort was made to minimize suffering. All of the animals were handled according to approved institutional animal care and use committee (IACUC) protocols (#10-062) of Rutgers University.

### Decision letter and Author response

Decision letter https://doi.org/10.7554/eLife.36697.027
Author response https://doi.org/10.7554/eLife.36697.028

## Additional files

### Supplementary files

• Transparent reporting form
DOI: https://doi.org/10.7554/eLife.36697.025

### Data availability

All data analysed during this study are included in the manuscript and supporting files. Source data values and images have been provided for Figure 1 and 2. All measured data is provided for Figures 3, 4, 5, 6, and 7, with individual measurement (including ~15,000 measurements for Fig 6).

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
