## [Decision Letter]

Thank you for submitting your article "Active presynaptic ribosomes in mammalian brain nerve terminals, and increased transmitter release after protein synthesis inhibition" for consideration by *eLife*. Your article has been reviewed by three peer reviewers, one of whom is a member of our Board of Reviewing Editors, and the evaluation has been overseen by Gary Westbrook as the Senior Editor. The following individual involved in review of your submission has agreed to reveal his identity: Henrique von Gersdorff (Reviewer #2). The reviewers have discussed the reviews with one another and the Reviewing Editor has drafted this decision to help you prepare a revised submission.

Summary:

In this study the authors use the calyx of Held synapse to demonstrate the functional impact of presynaptic protein synthesis on neurotransmitter release. The authors document active presynaptic ribosomes. Subsequently, using the protein synthesis inhibitor anisomycin, they show that acute inhibition of protein synthesis increases spontaneous release. The augmentation of spontaneous release rates could be occluded by prior tetanic stimulation. Interestingly, the same manipulation causes a reduction in paired pulse depression consistent with a decrease in release probability. This point is also verified by experiments assessing the released fraction of the total readily releasable pool. However, despite a decrease in initial release probability anisomycin treated synapses can sustain higher levels of release during repetitive stimulation. The authors posit that presynaptic protein synthesis at the calyx of Held suppresses spontaneous release and release during repetitive activity but maintains high levels of initial release probability. These findings are timely and add additional support to the increasing lines of evidence for presynaptic protein synthesis in mammalian central synapses. However, the author should address some key issues.

Essential revisions:

1) To further validate the anisomycin findings we encourage the authors to repeat one of the key experiments (increase in spontaneous release, decrease in initial Pr, increased sustained release) with another protein synthesis inhibitor to rule out potential unintended effects of the drug treatment. Importantly, there was strong consensus that the authors include a section in the discussion highlighting the need to further examine the mechanism of anisomycin's functional effects reported in the manuscript.

2) There seems to be a decrease in initial Pr during stimulus trains, but this was not due to a reduced initial RRP, but due to a smaller first EPSC amplitude (Figure 7A_1_ and B_1_). This last result appears to be inconsistent with Figure 5D, where a slightly smaller EPSC is observed for anisomycin, but this was not statistically significant. This may be an issue of series resistance (R_s_) changes in different data sets and the authors should double-check the data to make this internally consistent throughout the paper. Please report average R_s_ values in the methods for all conditions.

3) Throughout the manuscript, the authors do not address the existence of ribosomes in mitochondria, which are known to be present in presynaptic terminals. Although 5.8S rRNA antibodies should be able to locate cytoplasmic ribosomes and differentiate them from mitochondrial ones, the authors should specifically address in the manuscript the differences between cytoplasmic ribosomes and mitochondrial ribosomes, which are distinct both at the RNA and protein level. Considering the importance of this finding in relation to the main hypothesis raised by the authors (i.e., the localization of active cytoplasmic ribosomes to the presynaptic terminal), it appears to me that the specificity of their assays should be stated outright. This should not be a significant challenge for the authors.

4) Don't glial cells synthesize protein or contain ribosomes? Could the authors explain why these do not show up prominently in the staining (like SUnSET or 5.8S rRNA, which are not specific for neurons)?

5) The authors should discuss the relevance of an impressive set of studies by the laboratory of Noam Ziv which show that protein lifetime in neurons is significantly longer than initially anticipated. It is not only protein synthesis, but also control of protein degradation and turnover which could have a significant effect on synaptic protein composition.

---

## [Author Response]

Essential revisions:1) To further validate the anisomycin findings we encourage the authors to repeat one of the key experiments (increase in spontaneous release, decrease in initial Pr, increased sustained release) with another protein synthesis inhibitor to rule out potential unintended effects of the drug treatment. Importantly, there was strong consensus that the authors include a section in the discussion highlighting the need to further examine the mechanism of anisomycin's functional effects reported in the manuscript.

We fully agree with the reviewers’ main suggestion that testing an additional protein synthesis inhibitor would strengthen our findings on the effects of inhibiting protein synthesis. To address this, we repeated the main electrophysiology experiments using emetine, a compound that inhibits protein synthesis by a different mechanism than anisomycin as described in the revised manuscript. The results from these experiments now appear in a supplemental graph (Figure 6—figure supplement 1 and Figure 6—figure supplement 3; and Figure 7—figure supplement 1). For these experiments, we focused on the evoked release. In agreement with our findings following anisomycin treatment, we find:

1) emetine treatment increases the paired pulse ratio at both a 5 and 10 msec pulse interval (subsection “Paired pulse measurements indicate a presynaptic effect of translational inhibition”; Figure 6—figure supplement 1);

2) emetine treatment causes a decrease in depression during stimulation at 100 Hz and 200 Hz (Figure 6—figure supplement 3; and subsection “Paired pulse measurements indicate a presynaptic effect of translational inhibition”);

3) emetine treatment increases the relative vesicle replenishment rate compared to control recordings (subsection “Inhibiting protein synthesis affects vesicle release and replenishment” and Figure 7—figure supplement 1A, D);

4) emetine treatment decreased the initial probability of release for cells in which protein synthesis was inhibited by treatment with emetine (subsection “Inhibiting protein synthesis affects vesicle release and replenishment”; Figure 7—figure supplement 1 B, C, E, F).

5) emetine treatment also caused a reduction in the peak amplitude of synaptic responses in emetine treated neurons compared to controls (subsection “Paired pulse measurements indicate a presynaptic effect of translational inhibition”).

6) emetine treatment did not change the size of the RRP (subsection “Inhibiting protein synthesis affects vesicle release and replenishment”).

For the data analysis, we also used an additional method to measure the probability of release and the readily releasable pool at 100 Hz and 200 Hz (Figure 7—figure supplement 1C and 1F). Information on this analysis has been added to the Materials and methods section. To best determine the effects of emetine, we did one control recording and one recording from an emetine treated neuron from the same animal. This was done to help control for any variability in synaptic responses between different animals. The emetine treated and control recordings were pairwise tested. These experiments were performed approximately one year after the anisomycin experiments.

2) There seems to be a decrease in initial Pr during stimulus trains, but this was not due to a reduced initial RRP, but due to a smaller first EPSC amplitude (Figure 7A_1_ and B_1_). This last result appears to be inconsistent with Figure 5D, where a slightly smaller EPSC is observed for anisomycin, but this was not statistically significant. This may be an issue of series resistance (R_s_) changes in different data sets and the authors should double-check the data to make this internally consistent throughout the paper. Please report average R_s_ values in the methods for all conditions.

The reviewers have emphasized the importance of low and stable series resistance values and the potential difficulties of comparing different data sets. We agree that low and stable series resistance values are necessary to accurately measure the peak amplitudes, and the kinetics of the postsynaptic response. We note that the recordings used in Figure 5D are the same recordings used in Figure 6 and Figure 7, but also contain several additional recordings. These additional recordings were not used in Figure 6 and Figure 7 for reasons that include brief recording durations, response failures, and inability to maintain series resistance compensation throughout the recording. These additional recordings were therefore not included in the analysis of the repetitive stimulation experiments (Figure 6 and Figure 7). Accordingly, we have now applied the same criteria to the recordings shown in Figure 5, such that the same set of highly stable recordings are used in Figure 5, Figure 6 and Figure 7. We now state this in the Results section.

To further address the series resistance concerns, we constructed graphs to show the average series resistance throughout each recording, under control and anisomycin treated conditions (Figure 6—figure supplement 4A, B). To show the relationship between series resistance and the peak amplitude, we also show the cell average of the first EPSC for the 100 Hz tetanus recordings along with the corresponding series resistance for each recording (Figure 6—figure supplement 4C, D). The same is shown for the 200 Hz data (Figure 6—figure supplement 4E, F).

In summary, the data in Figure 5 can now be directly compared to the data in Figure 6 and Figure 7, and the series resistance values for the 100 and 200 Hz repetitive stimulation data are now shown. In addition, we have included the following information in the Materials and methods section:

“Postsynaptic series resistances (R_s_) for voltage clamp recordings was less than 15 MΩ and typically varied less than 2MΩ throughout the recording. For the recordings that appear in Figure 5, Figure 6 and Figure 7, showing EPSC response properties in control and after anisomycin treatment, the average R_s_ values and corresponding EPSC amplitudes in response to stimulation at 100 Hz and 200 Hz are graphed (see Figure 6—figure supplement 4A, B). We also provide a graph of R_s_ values and corresponding EPSC peak amplitude for the data in Figure 6 and Figure 7 (see Figure 6—figure supplement 4C,D,E,F).”

3) Throughout the manuscript, the authors do not address the existence of ribosomes in mitochondria, which are known to be present in presynaptic terminals. Although 5.8S rRNA antibodies should be able to locate cytoplasmic ribosomes and differentiate them from mitochondrial ones, the authors should specifically address in the manuscript the differences between cytoplasmic ribosomes and mitochondrial ribosomes, which are distinct both at the RNA and protein level. Considering the importance of this finding in relation to the main hypothesis raised by the authors (i.e., the localization of active cytoplasmic ribosomes to the presynaptic terminal), it appears to me that the specificity of their assays should be stated outright. This should not be a significant challenge for the authors.

We thank the reviewers for pointing out the need to clearly state that the antibody against 5.8S rRNA is eukaryotic specific. This is an important point and critical for interpretation of our data. The revised manuscript (subsection “Evidence for presynaptic ribosomes at the calyx of Held nerve terminal”) now states the following: “To investigate presynaptic protein synthesis, we first determined whether 5.8S ribosomal RNA (rRNA), a major component of the eukaryotic ribosome which is required to execute ribosomal translocation (Lerner et al., 1981; Abou, Elela and Nazar, 1997; Koenig et al., 2000), is present in the nerve terminal. This component is eukaryotic specific and part of the large 80S ribosomal subunit. Therefore, the antibody does not label mitochondrial ribosomes which are more prokaryotic in their composition.”

Concerning the puromycin assay, we do not know of any studies measuring mitochondrial protein synthesis with the SUnSET method, so it’s not clear if it is possible to measure active mitochondrial protein synthesis, particularly in situ, using the SUnSET assay. In addition, to test the specificity of the SUnSET assay on protein synthesis, we used anisomycin which specifically inhibits eukaryotic protein synthesis. We note that anisomycin can be used to inhibit cellular protein synthesis to allow the specific study of mitochondrial protein synthesis in mammalian cells. Therefore, if mitochondrial protein synthesis contributed to the fluorescent signal in the SUnSET assay, the anisomycin treatment would not eliminate the fluorescence signal that shows active protein synthesis. This is now explained in subsection “Functional presynaptic ribosomes” of the revised manuscript.

4) Don't glial cells synthesize protein or contain ribosomes? Could the authors explain why these do not show up prominently in the staining (like SUnSET or 5.8S rRNA, which are not specific for neurons)?

This is a very good point. We consistently observed non-calyceal 5.8S rRNA and SUnSET signal that is presumably coming from other neurons and/or glia. In Figure 1 and Figure 2, there are clear regions that contain ribosomal signal (or puromycin signal) that are not localized with VGLUT1. To clearly demonstrate that the protein synthesis signal is coming from the calyx compartment, we chose examples where the relative signal in the calyx was higher than the signal in the adjacent regions. In these examples, shown in Figure 1 and Figure 2, the location of ribosomes, or ribosomal activity in the presynaptic terminal was presumably higher than the level in adjacent glial cells. In addition, the adjacent glia in certain areas could be damaged, absent or, as stated above, simply have a low signal relative to the calyx in those specific areas. For example, in Figure 1C, we show a presynaptic signal in the absence of a postsynaptic signal.

To further clarify and emphasize this, we have added the following sentence (subsection “Functional presynaptic ribosomes”): “This serves to further demonstrate an isolated presynaptic signal. Localizing puromycin fluorescence to the presynaptic terminal demonstrates the presence of active presynaptic ribosomes.”

We have now pointed out that there are adjacent regions with high background activity in Figure 1 (subsection “Evidence for presynaptic ribosomes at the calyx of Held nerve terminal”). We also clarified our statement on the average intensity ratio of presynaptic to background 5.8s ribosomal signal, noting that this ratio refers to the region shown in Figure 1C and D (subsection “Evidence for presynaptic ribosomes at the calyx of Held nerve terminal”). In addition, we have added a supplemental figure (Figure 2—figure supplement 1). Here, we enlarged the images in Figure 2A, and show a higher resolution section of these images to demonstrate that cells neighboring the calyx of Held can show high basal levels of activity (subsection “Functional presynaptic ribosomes”).

5) The authors should discuss the relevance of an impressive set of studies by the laboratory of Noam Ziv which show that protein lifetime in neurons is significantly longer than initially anticipated. It is not only protein synthesis, but also control of protein degradation and turnover which could have a significant effect on synaptic protein composition.

This is important to discuss in relation for the need of local translation to maintain synaptic activity at the calyx of Held synapse. We have added a new section to discuss Dr. Ziv’s findings, and how they may relate to our findings (Discussion section).